# TOWARDS UNDERSTANDING NEURAL COLLAPSE: THE EFFECTS OF BATCH NORMALIZATION AND WEIGHT DECAY

## ABSTRACT

Neural Collapse ($\mathcal{NC}$) is a geometric structure recently observed in the final layer of neural network classifiers. In this paper, we investigate the interrelationships between batch normalization (BN), weight decay, and proximity to the $\mathcal{NC}$ structure. Our work investigates the geometrically intuitive intra-class and inter-class cosine similarity measure, which encapsulates multiple core aspects of $\mathcal{NC}$. Leveraging this measure, we establish theoretical guarantees for the emergence of $\mathcal{NC}$ under the influence of last-layer BN and weight decay, specifically in scenarios where the regularized cross-entropy loss is near-optimal. Experimental evidence substantiates our theoretical findings, revealing a pronounced occurrence of $\mathcal{NC}$ in models incorporating BN and appropriate weight-decay values. This combination of theoretical and empirical insights suggests a greatly influential role of BN and weight decay in the emergence of $\mathcal{NC}$.

## 1 INTRODUCTION

Over the past decade, deep learning and neural networks have revolutionized the field of machine learning and artificial intelligence, enabling machines to perform complex tasks previously thought to be beyond their capabilities. However, despite tremendous empirical advances, a comprehensive theoretical and mathematical understanding of the success behind neural networks, even for the simplest types, is still unsatisfactory. Analyzing Neural Networks using traditional statistical learning theory has encountered significant difficulties due to the high level of non-convexity, over-parameterization, and optimization-dependent properties.

Papyan et al. (2020) recently empirically observed an elegant mathematical structure in multiple successful neural network-based visual classifiers and named the phenomenon "Neural Collapse" (abbreviated $\mathcal{NC}$ in this work). Specifically, $\mathcal{NC}$ is a geometric structure of the learned last-layer/penultimate-layer feature and weights at the terminal phase of deep neural network training. Neural Collapse states that after sufficient training of successful neural networks: **NC1)** The intra-class variability of the last-layer feature vectors tends to zero *(Variability Collapse)*; **NC2)** The mean class feature vectors become equal-norm and forms a Simplex Equiangular Tight Frame (ETF) around the center up to re-scaling.; *(Convergence to Simplex ETF)* **NC3)** The last layer weight vectors converge to match the feature class means up to re-scaling *(Self-Duality)*; **NC4)** The last layer of the network behaves the same as a "Nearest Class Center" decision rule *(Convergence to NCC)*

Notably, an Equiangular Tight Frame (ETF) is a set of vectors in a high-dimensional space that are evenly spaced from each other, such that they form equal angles with one another and are optimally arranged for maximal separability. In our context of $\mathcal{NC}$, a simplex Equiangular Tight Frame in Euclidean space is defined as follows:

**Definition 1.1** (Simplex ETF, Papyan et al. (2020))**.** *A simplex ETF is a collection of $C$ points in $\mathbb{R}^d$ specified by the columns of*

$$M^{\star} = \alpha U \left( I_C - \frac{1}{C} \mathbf{1}_C \mathbf{1}_C^{\top} \right).$$

*where $\alpha \in \mathbb{R}^+$ and $U \in \mathbb{R}^{d \times C}$ is a partially orthogonal matrix ($U^{\top} U = I$).*

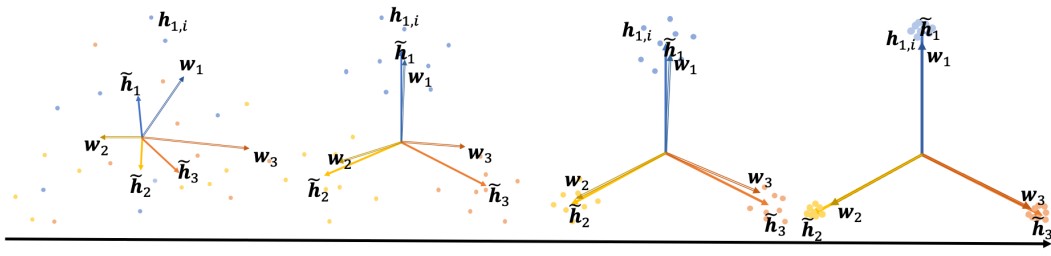

Training Time

Figure 1: Visualization of $\mathcal{NC}$ (Papyan et al. (2020)). We use an example of three classes and denote the last-layer features $\mathbf{h}_{c,i}$, mean class features $\tilde{\mathbf{h}}_c$, and last-layer class weight vectors $\mathbf{w}_{c,i}$. Circles denote individual last-layer features, while compound and filled arrows denote class weight and mean feature vectors, respectively. As training progresses, the last-layer features of each class collapse to their corresponding class means (NC1), different class means converge to the vertices of the simplex ETF (NC2), and the class weight vector of the last-layer linear classifier approaches the corresponding class means (NC3).

These observations of Neural Collapse reveal compelling insights into the symmetry and mathematical preferences of over-parameterized neural network classifiers. Intuitively, the last-layer features acquire the most suitable geometric feature representation for their specific classification task that maximizes inter-class separation while simultaneously discarding information about variations within individual classes. Subsequently, further work has demonstrated that Neural Collapse may play a significant role in the generalization, transfer learning (Galanti et al. (2022b)), depth minimization (Galanti et al. (2022a)), and implicit bias of neural networks (Poggio & Liao (2020)). Additionally, insights provided by Neural Collapse have been a powerful tool in exploring the intermediate layers of neural network classifiers and representations learned by self-supervised learning models.( Ben-Shaul et al. (2023); Ben-Shaul & Dekel (2022))

## 1.1 OUR CONTRIBUTIONS

In this paper, we theoretically and empirically investigate the question:

**What is a minimal set of conditions that would guarantee the emergence of $\mathcal{NC}$?**

Our results show that batch normalization, large weight decay, and near-optimal cross-entropy loss are sufficient conditions for several core properties of $\mathcal{NC}$ and $\mathcal{NC}$ is most significant when all these conditions are satisfied. Specifically, we provide the following contributions:

- We quantitatively investigate the intra-class and inter-class cosine similarity measure, a simple and geometrically intuitive quantity that measures the proximity of a set of feature vectors to several core structural properties of $\mathcal{NC}$. (Section 2.2)

- Under the cosine similarity measure, we show a theoretical guarantee of the proximity to $\mathcal{NC}$ for any neural network classifier without bias terms with near-optimal regularized cross-entropy loss, batch-normalized last-layer feature vectors, and last-layer weight decay. (Theorem 2.2)

- Our empirical evidence shows that $\mathcal{NC}$ is most significant with both batch normalization and high weight decay values under the cosine similarity measure. (Section 3)

Combining our theoretical and empirical results, we conclude that batch normalization along with weight decay may be greatly influential conditions for the emergence of $\mathcal{NC}$.

## 1.2 RELATED THEORETICAL WORKS ON THE EMERGENCE OF NEURAL COLLAPSE

The empirical $\mathcal{NC}$ phenomenon has inspired a recent line of work to theoretically investigate its emergence under different settings. Several studies have focused on the unconstrained features model or layer-peeled model, first introduced by Mixon et al. (2020), where the last layer features are

treated as free optimization variables. Such simplification is based on the observation that most modern neural networks are highly over-parameterized and are capable of learning any feature representations. Following this model, several works have demonstrated that solutions satisfying Neural Collapse are the only global optimizers under both CE (Ji et al. (2022); Zhu et al. (2021); Lu & Steinerberger (2022)) and MSE loss (Han et al. (2022); Zhou et al. (2022)) under different settings such as regularization and normalization. Recent works have also focused on analyzing the unconstrained features model's gradient dynamics and optimization landscape (Mixon et al. (2020); Zhu et al. (2021); Ji et al. (2022); Han et al. (2022); Yaras et al. (2022)). Collectively, these works establish that, under both CE and MSE loss, the unconstrained features model has a benign global optimization landscape where every local minima solution satisfies the Neural Collapse structure and other critical points are strict saddle points with negative curvature. Furthermore, following the gradient flow or first-order optimization method would lead to solutions satisfying the Neural Collapse structure. Although works have been done in an idealized setting where gradient-based optimization is performed directly on the last layer features, it should be noted that this assumption is unrealistic. Optimizing the weights in earlier layers can have a significantly different effect from directly optimizing the last-layer features, even in over-parameterized networks. Besides the layer-peeled model, Poggio & Liao (2020) have demonstrated the Neural Collapse structure for binary classification when each individual sample achieves zero gradients with MSE loss, while Tirer & Bruna (2022) and Súkeník et al. (2023) extends the analysis under MSE loss to deeper models.

For a table comparing the model and contributions of prior work theoretically investigating the emergence of $\mathcal{NC}$, see Appendix section C.

**Our Work: $\mathcal{NC}$ Proximity Under Near-optimal Loss**  Building on the layer-peeled model from prior research, our theoretical approach offers a unique perspective, focusing on the *near-optimal regime* and avoiding less realistic assumptions of achieving exact optimal loss and directly optimizing the last-layer feature vectors. Our approach provides further insights into $\mathcal{NC}$ in realistic neural network training as 1) the near-optimal regime is often more reflective of the realities of neural network training, with the theoretical optimal loss often being unattainable in practice; 2) in contrast to landscape or gradient flow analyses on the layer-peeled model, our findings are optimization-agnostic and applicable in practical scenarios where direct optimization of the last-layer features is unfeasible; 3) our emphasis on measuring the *proximity* to $\mathcal{NC}$, rather than achieving exact $\mathcal{NC}$, unveils additional insights, especially in instances where exact $\mathcal{NC}$ is unattainable.

## 2 THEORETICAL RESULTS

### 2.1 PROBLEM SETUP AND NOTATIONS

**Neural Network with Cross-Entropy Loss.**  In this work, we consider neural network classifiers without bias terms trained using cross-entropy loss functions on a balanced dataset. A vanilla deep neural network classifier is composed of a feature representation function $\phi_{\boldsymbol{\theta}}(\boldsymbol{x})$ and a linear classifier parameterized by $\mathbf{W}$. Specifically, a $L$-layer vanilla deep neural network can be mathematically formulated as:

$$f(\boldsymbol{x}; \boldsymbol{\theta}) = \underbrace{\boldsymbol{W}^{(L)}}_{\text{Last layer weight } \mathbf{W} = \mathbf{W}^{(L)}} \underbrace{BN\left(\sigma\left(\boldsymbol{W}^{(L-1)}\cdots\sigma\left(\boldsymbol{W}^{(1)}\boldsymbol{x} + \boldsymbol{b}^{(1)}\right) + \cdots + \boldsymbol{b}^{(L-1)}\right)\right)}_{\text{last-layer feature } \boldsymbol{h} = \phi_{\boldsymbol{\theta}}(\boldsymbol{x})}.$$

Each layer is composed of an affine transformation parameterized by weight matrix $\boldsymbol{W}^{(l)}$ and bias $\boldsymbol{v}^{(l)}$ followed by a non-linear activation $\sigma$ which may contain element-wise transformation such as $\text{ReLU}(x) = \max\{x, 0\}$ as well as normalization techniques such as batch normalization.

The network is trained by minimizing the empirical risk over all samples $\{(\boldsymbol{x}_{c,i}, \boldsymbol{y}_c)\}, c \in [C], i \in [N]$ where each class contains $N$ samples and $\boldsymbol{y}_c$ is the one-hot encoded label vector for class $c$. We also denote $\mathbf{h}_{c,i} = \phi_{\boldsymbol{\theta}}(\boldsymbol{x}_{c,i})$ as the last-layer feature corresponding to $\boldsymbol{x}_{c,i}$. The training process minimizes the average cross-entropy loss

$$\mathcal{L} = \frac{1}{CN}\sum_{c=1}^{C}\sum_{i=1}^{N}\mathcal{L}_{\text{CE}}\left(f(\boldsymbol{x}_{c,i}; \boldsymbol{\theta}), \boldsymbol{y}_c\right) = \frac{1}{CN}\sum_{c=1}^{C}\sum_{i=1}^{N}\mathcal{L}_{\text{CE}}\left(\boldsymbol{W}\boldsymbol{h}_{c,i}, \boldsymbol{y}_c\right),$$

where the cross entropy loss function for a one-hot encoding $\boldsymbol{y}_c$ is:

$$\mathcal{L}_{\mathrm{CE}}(\boldsymbol{z}, \boldsymbol{y}_c) = -\log\left(\frac{\exp(z_c)}{\sum_{c'=1}^{C} \exp(z'_c)}\right).$$

**Batch Normalization and Weight Decay.** For a given batch of vectors $\mathbf{v}_1, \mathbf{v}_2, \cdots \mathbf{v}_b \subset \mathbb{R}^d$, let $v^{(k)}$ denote the $k$'th element of $\mathbf{v}$. Batch Normalization (BN) developed by Ioffe & Szegedy (2015) performs the following operation along each dimension $k \in [d]$:

$$BN(\mathbf{v}_i)^{(k)} = \frac{v_i^{(k)} - \mu^{(k)}}{\sigma^{(k)}} \times \gamma^{(k)} + b^{(k)}.$$

Where $\mu^{(k)}$ and $(\sigma^{(k)})^2$ are the mean and variance along the $k$'th dimension of all the vectors in the batch. The vectors $\boldsymbol{\gamma}$ and $\boldsymbol{b}$ are trainable parameters that represent the desired variance and mean after BN. BN has been empirically demonstrated to facilitate convergence and generalization and is adopted in many popular network architectures. In our work, we consider BN layers without bias (i.e. $\boldsymbol{b} = 0$)

Weight decay is a technique in deep learning training that facilitates generalization by penalizing large weight vectors. Specifically, the Frobenius norm of each weight matrix $\boldsymbol{W}^{(l)}$ and batch normalization weight vector $\boldsymbol{\gamma}^{(l)}$ is added as a penalty term to the final cross-entropy loss. Thus, the final loss function with weight decay parameter $\lambda$ is

$$\mathcal{L}_{\mathrm{reg}} = \mathcal{L} + \frac{\lambda}{2} \sum_{l=1}^{L} (\|\boldsymbol{\gamma}^{(l)}\|^2 + \|\mathbf{W}^{(l)}\|_F^2),$$

where $\gamma^{(l)} = 0$ for layers without batch normalization. In our theoretical analysis, we consider the simplified layer-peeled model that only applies weight decay on the network's final linear and batch normalization layer. Under this setting, the final regularized loss is:

$$\mathcal{L}_{\mathrm{reg}} = \mathcal{L} + \frac{\lambda}{2} (\|\boldsymbol{\gamma}\|^2 + \|\mathbf{W}\|_F^2),$$

where $\mathbf{W}$ is the last layer weight matrix and $\boldsymbol{\gamma} = \boldsymbol{\gamma}^{(L-1)}$ is the weight of the batch normalization layer before the final linear transformation.

## 2.2 Cosine Similarity Measure of Neural Collapse

Numerous measures of NC have been used in past literature, including within-class covariance (Papyan et al. (2020)), signal-to-noise (SNR) ratio (Han et al. (2022)), as well as class distance normalized variance (CDNV, Galanti et al. (2022b)). While these measures all indicate the emergence of $\mathcal{NC}$ when the measured value approaches zero and provides convergence guarantees to Neural Collapse, they do not provide a *geometrically* straightforward and intuitive measure of how close a given structure is to $\mathcal{NC}$ when the values are non-zero.

In this work, we investigate the cosine similarity measure as a measure (Kornblith et al. (2020)) of $\mathcal{NC}$, which focuses on simplicity and geometric interpretability at the cost of discarding norm information. We note that cosine similarity is also widely used as a measure of similarity between features of different samples in both practical feature learning and machine learning theory, which makes our results more relevant in these fields.

For a given class $c$, the average intra-class cosine similarity for class $c$ is defined as the average cosine similarity of picking two feature vectors in the class after centering with respect to the global mean feature vector $\tilde{\mathbf{h}}_G$:

$$intra_c = \frac{1}{N^2} \sum_{i=1}^{N} \sum_{j=1}^{N} \cos_\angle(\mathbf{h}_{c,i} - \tilde{\mathbf{h}}_G, \mathbf{h}_{c,j} - \tilde{\mathbf{h}}_G),$$

where

$$\cos_\angle(\mathbf{x}, \mathbf{y}) = \frac{\mathbf{x}^\mathsf{T} \mathbf{y}}{\|\mathbf{x}\| \cdot \|\mathbf{y}\|}.$$

is the vector cosine similarity measure. Similarity, the inter-class cosine similarity between two classes $c, c'$ is defined as the average cosine similarity of picking one feature vector of class $c$ and another from class $c'$:

$$inter_{c,c'} = \frac{1}{N^2} \sum_{i=1}^{N} \sum_{j=1}^{N} \cos_\angle(\mathbf{h}_{c,i} - \tilde{\mathbf{h}}_G, \mathbf{h}_{c',j} - \tilde{\mathbf{h}}_G)$$

We note that since the final layer is a batch normalization layer without the bias term, the global mean is guaranteed to be zero, and thus the global mean can be discarded. In the remainder of this work we ignore the $\tilde{\mathbf{h}}_G$) unless otherwise stated.

**Relationship with $\mathcal{NC}$**  While cosine-similarity does not measure the degree of norm equality, it can describe *necessary* conditions for the core observations of $\mathcal{NC}$ as follows:

(NC1) *(Variability Collapse)* NC1 implies that all features in the same class collapse to the class mean and have the same vector value. Therefore, all features in the same class must be in the same direction and achieve an intra-class cosine similarity $intra_c = 1$.

(NC2) *(Convergence to Simplex ETF)* NC2 implies that class means converge to the vertices of a simplex ETF. Combined with NC1, this implies that the angle between every pair of features from different classes must be $-\frac{1}{C-1}$ (a property of the simplex ETF over $C$ points). Therefore, the inter-class cosine similarity between each pair of classes must be $inter_{c,c'} = -\frac{1}{C-1}$

With the above problem formulation, we now present our main theorems for $\mathcal{NC}$ in neural network classifiers with near-optimal training cross-entropy loss. Before presenting our core theoretical result on batch normalization and weight decay, we first present a more general preliminary theorem that provides theoretical bounds for the intra-class and inter-class cosine similarity for any classifier with near-optimal (unregularized) average cross-entropy loss.

## 2.3 Main Results

Our first theorem states that if the average last-layer feature norm and the last-layer weight matrix norm are both *bounded*, then achieving *near-optimal loss* implies that *most classes* have intra-class cosine similarity near one and *most pairs of classes* have inter-class cosine similarity near $-\frac{1}{C-1}$. The non-asymptotic version of the theorems and similar results for NC3 under the cosine similarity measure is in the appendix.

**Theorem 2.1** ($\mathcal{NC}$ proximity guarantee with bounded norms). *For any neural network classifier without bias terms trained on a dataset with the number of classes $C \geq 3$ and samples per class $N \geq 1$, and the last layer feature dimension $d \geq C$, under the following assumptions:*

*1. The quadratic average of the last-layer feature norms $\sqrt{\frac{1}{CN} \sum_{c=1}^{C} \sum_{i=1}^{N} \|\mathbf{h}_{c,i}\|^2} \leq \alpha$*

*2. The Frobenius norm of the last-layer weight $\|\mathbf{W}\|_F \leq \sqrt{C}\beta$*

*3. The average cross-entropy loss over all samples $\mathcal{L} \leq m + \epsilon$ for small $\epsilon > 0$*

*where $m = \log(1 + (C-1)\exp(-\frac{C}{C-1}\alpha\beta))$ is the minimum achievable loss for any set of weight and feature vectors satisfying the norm constraints, then for at least $1 - \delta$ fraction of all classes , with $\frac{\epsilon}{\delta} \ll 1$, there is*

$$intra_c \geq 1 - O\left(\frac{e^{O(C\alpha\beta)}}{\alpha\beta}\sqrt{\frac{\epsilon}{\delta}}\right),$$

*and for at least $1 - \delta$ fraction of all pairs of classes $c, c'$, with $\frac{\epsilon}{\delta} \ll 1$, there is*

$$inter_{c,c'} \leq -\frac{1}{C-1} + O\left(\frac{e^{O(C\alpha\beta)}}{\alpha\beta}(\frac{\epsilon}{\delta})^{1/6}\right).$$

**Remarks.**

- We only consider the near-optimal regime where $\epsilon \ll 1$. However, a near-optimal cross-entropy training loss is demonstrated in most successful neural network classifiers exhibiting $\mathcal{NC}$, including all the original experiments by Papyan et al. (2020), at the terminal phase of training.

- Since $\frac{e^{O(C\alpha\beta)}}{\alpha\beta}$ is a mostly increasing function of $\alpha\beta$, lower last-layer feature and weight norms can provide stronger guarantees on Neural Collapse measured using cosine similarity.

*Proof Sketch of Theorem 2.1.* Our proof is inspired by the optimal-case proof of Lu & Steinerberger (2022), which shows the global optimality conditions using Jensen's inequality. Our core lemma shows that if a set of variables achieves roughly equal value on the LHS and RHS of Jensen's inequality for a strongly convex function (such as $\exp(x)$), then the mean of every subset cannot deviate too far from the global mean:

**Lemma 2.1** (Subset mean close to global mean by Jensen's inequality on strongly convex functions)**.** *Let $\{x_i\}_{i=1}^N \subset \mathcal{I}$ be a set of $N$ real numbers, let $\tilde{x} = \frac{1}{N} \sum_{i=1}^N x_i$ be the mean over all $x_i$ and $f$ be a function that is $m$-strongly-convex on $\mathcal{I}$. If*

$$\frac{1}{N} \sum_{i=1}^N f(x_i) \leq f(\tilde{x}) + \epsilon,$$

*i.e., Jensen's inequality is satisfied with gap $\epsilon$, then for any subset of samples $S \subseteq [N]$, let $\delta = \frac{|S|}{N}$, there is*

$$\tilde{x} + \sqrt{\frac{2\epsilon(1-\delta)}{m\delta}} \geq \frac{1}{|S|} \sum_{i \in S} x_i \geq \tilde{x} - \sqrt{\frac{2\epsilon(1-\delta)}{m\delta}}.$$

This lemma can serve as a general tool to convert optimal-case conditions derived using Jensen's inequality into high-probability proximity bounds under near-optimal conditions.

Using the strong convexity of $\exp(x)$ and $\log(1 + (C-1)\exp(x))$ along with Lemma 2.1 and the optimal case proof of Lu & Steinerberger (2022), we show that most classes $c$ much have high same-class weight-feature vector cosine similarity, and most pairs of classes $c, c'$ have inter-class weight-feature vector cosine similarity. This upper and lower bound is then used to lower bound $\|\tilde{\tilde{\mathbf{h}}}_c\|$ and upper bound $\langle \tilde{\tilde{\mathbf{h}}}_c, \tilde{\tilde{\mathbf{h}}}_{c'} \rangle$ where

$$\tilde{\tilde{\mathbf{h}}}_c = \frac{1}{N} \sum_{i=1}^N \frac{\mathbf{h}_{c,i}}{\|\mathbf{h}_{c,i}\|}$$

is the mean *normalized* feature vector of class $c$. The intra-class and inter-class cosine similarity follows immediately from these results.

$\square$

Our preliminary theorem above shows that lower values of the average feature norm and weight Frobenius norm of the final layer provide stronger guarantees of the proximity to $\mathcal{NC}$. Note that weight decay is used to regularize the norms of weight matrices and weight vectors. Therefore, higher weight decay values should result in smaller weight matrix and weight vector norms. Our following proposition shows that regularizing the weight vector of a batch normalization layer without the bias term is equivalent to regularizing the quadratic average of the feature norms of its output vectors:

**Proposition 2.1** (BN normalizes quadratic average of feature norms)**.** *Let $\{\mathbf{h}_i\}_{i=1}^B$ be a set of Batch Normalized feature vectors with variance vector $\boldsymbol{\gamma}$ and bias term $\boldsymbol{\beta} = 0$ (i.e. $\mathbf{h}_i = BN(\mathbf{x}_i)$ for some $\{\mathbf{x}_i\}_{i=1}^B$). Then*

$$\sqrt{\frac{1}{N} \sum_{i=1}^N \|\mathbf{h}_i\|_2^2} = \|\boldsymbol{\gamma}\|_2.$$

Therefore, regularizing the batch normalization variance vector is effectively equivalent to regularizing the quadratic average of the feature norms. Intuitively, under the same other conditions, a higher regularization coefficient in the training loss function should result in lower values of the regularized parameters. Therefore, a higher weight decay value (i.e., regularization coefficient of the weight matrices and variance vectors) should result in a lower weight norm and last-layer feature norm and a tighter bound in Theorem 2.1. This intuition is formalized in the following main theorem:

**Theorem 2.2** ($\mathcal{NC}$ proximity guarantee with layer-peeled BN and WD). *For a neural network classifier without bias terms trained on a dataset with the number of classes $C \geq 3$ and samples per class $N \geq 1$, and the last layer feature dimension $d \geq C$, under the following assumptions:*

1. *The network contains a batch normalization layer without bias term before the final layer with trainable weight vector $\boldsymbol{\gamma}$;*

2. *The layer-peeled regularized cross-entropy loss with weight decay $\lambda$*

$$\mathcal{L}_{\text{reg}} = \frac{1}{CN} \sum_{c=1}^{C} \sum_{i=1}^{N} \mathcal{L}_{\text{CE}} \left( \boldsymbol{W} \boldsymbol{h}_{c,i}, \boldsymbol{y}_c \right) + \frac{\lambda}{2} (\|\boldsymbol{\gamma}\|^2 + \|\mathbf{W}\|_F^2)$$

*satisfies $\mathcal{L}_{\text{reg}} \leq m_{\text{reg}} + \epsilon$ for small $\epsilon$; where $m_{reg}$ is the minimum achievable regularized loss*

*then for at least $1 - \delta$ fraction of all classes , with $\frac{\epsilon}{\delta} \ll 1$, there is*

$$intra_c \geq 1 - O\left( (C/\lambda)^{O(C)} \sqrt{\frac{\epsilon}{\delta}} \right),$$

*and for at least $1 - \delta$ fraction of all pairs of classes $c, c'$, with $\frac{\epsilon}{\delta} \ll 1$, there is*

$$inter_{c,c'} \leq -\frac{1}{C-1} + O\left( (C/\lambda)^{O(C)} (\frac{\epsilon}{\delta})^{1/6} \right).$$

Since $e^{O(C/\lambda)}$ is an decreasing function of $\lambda$, higher values of $\lambda$ would result in smaller values of both $O(e^{O(C/\lambda)}(\frac{\epsilon}{\delta})^{1/6})$ and $O(e^{O(C/\lambda)}\sqrt{\frac{\epsilon}{\delta}})$. As such, under the presence of batch normalization and weight decay of the final layer, larger values of weight decay provide stronger $\mathcal{NC}$ guarantees in the sense that the intra-class cosine similarity of most classes is nearer to 1 and the inter-class cosine similarity of most pairs of classes is nearer to $-\frac{1}{C-1}$

### 2.4 CONCLUSION

Our theoretical result shows that last-layer BN, last-layer weight decay, and near-optimal average cross-entropy loss are sufficient conditions to guarantee proximity to the $\mathcal{NC}$ structure as measured using cosine similarity, regardless of the training method and earlier layer structure. Moreover, such a guarantee is optimization-independent

## 3 EMPIRICAL RESULTS

In this chapter, we present empirical evidence on the importance of batch normalization and weight decay on the emergence of Neural Collapse. Specifically, we compare the emergence of Neural Collapse in terms of the minimum intra-class cosine similarity over all classes and maximum inter-class cosine similarity over all pairs of classes. Our experiments show that **models with batch normalization and appropriate weight decay achieve the highest levels of $\mathcal{NC}$ measured using cosine similarity**, which supports the predictions of Theorem 2.2.

### 3.1 EXPERIMENTS WITH SYNTHETIC DATASETS

Our first set of experiments considers the simple setting of using a vanilla neural network (i.e., Multi-Layer Perceptron) to classify a well-defined synthetic dataset of different separation complexities. We aim to use straightforward model architectures and well-defined datasets of different complexities to explore the effect of different hyperparameters in $\mathcal{NC}$ under a controlled setting.

For datasets, we consider two different datasets of increasing classification difficulty: 1) The 4-class conic hull dataset, where two intersecting hyperplanes separate the input space into four classes; 2) the MLP3 dataset, where class labels are generated by the predicted labels of a 3-layer Neural Network with randomly generated weights. In the appendix, we also provide results for MLP6 and MLP9 datasets, created in a similar manner but for 6 and 9-layer neural networks.

The models used in the experiments are 4-layer and 6-layer multi-layer perception (MLP) models with ReLU activation. We compare models with and without batch normalization, where the batch-normalized models have batch-normalization layers between every adjacent linear layer. We train each model on the same synthetic dataset with 8000 training samples over 15 weight decay values ranging from 0.0001 to 0.1. For each experiment, we record the average and **minimum** intra-class cosine similarity among all classes and the average and **maximum** inter-class cosine similarity among all pairs of classes as defined in section 2.2. Each model is trained for 300 epochs using the Adam optimizer.

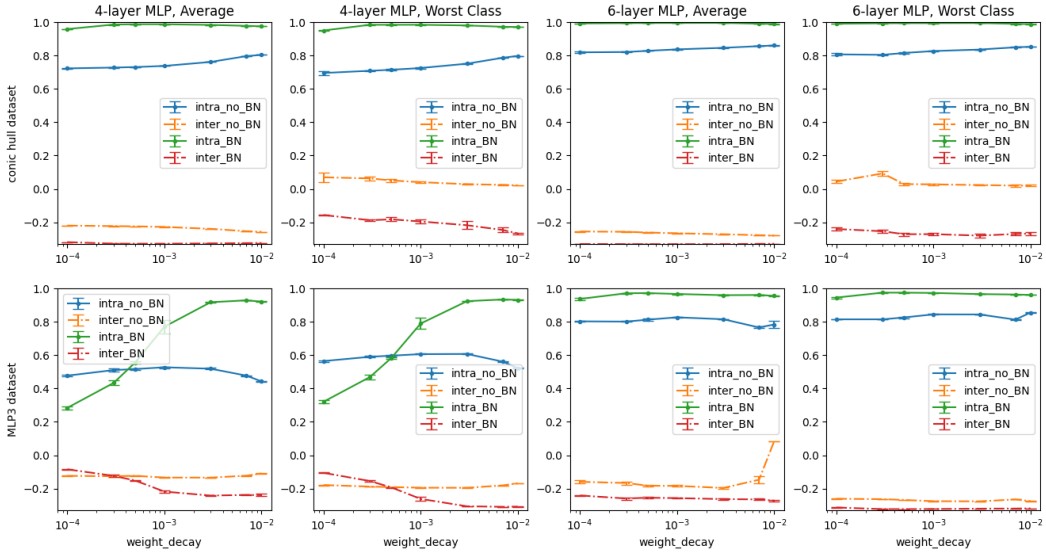

Figure 2: Minimum intra-class and maximum inter-class Cosine Similarity for 4-layer and 6-layer MLP under Different WD and BN. Higher values of intra-class and lower values of inter-class cosine similarity imply a higher degree of Neural Collapse. Both the average and worst measures over classes are presented. Error bars refer to the standard deviation over five different experiments.

### 3.2 EXPERIMENT WITH REAL-WORLD DATASETS

Our next set of experiments explores the effect of Batch Normalization and Weight Decay using standard computer vision datasets MNIST (LeCun et al. (2010)) and CIFAR-10 (Krizhevsky (2009)). Specifically, we explore the difference in the degree of Neural Collapse between convolutional neural network architectures with and without Batch Normalization across different weight decay parameters. Notably, we compare the results of 2 different implementations of the VGG11 and VGG19 (Simonyan & Zisserman (2015)) convolutional neural network, one of which applies batch normalization after each convolution layer. Results are presented in Figure 3

### 3.3 CONCLUSION

Our experiments show that, in both synthetic and realistic scenarios, the highest level of $\mathcal{NC}$ is achieved by models with BN and appropriate weight decay. Moreover, BN allows the degree of $\mathcal{NC}$ to increase smoothly along with the increase of weight decay within the range of perfect interpolation,

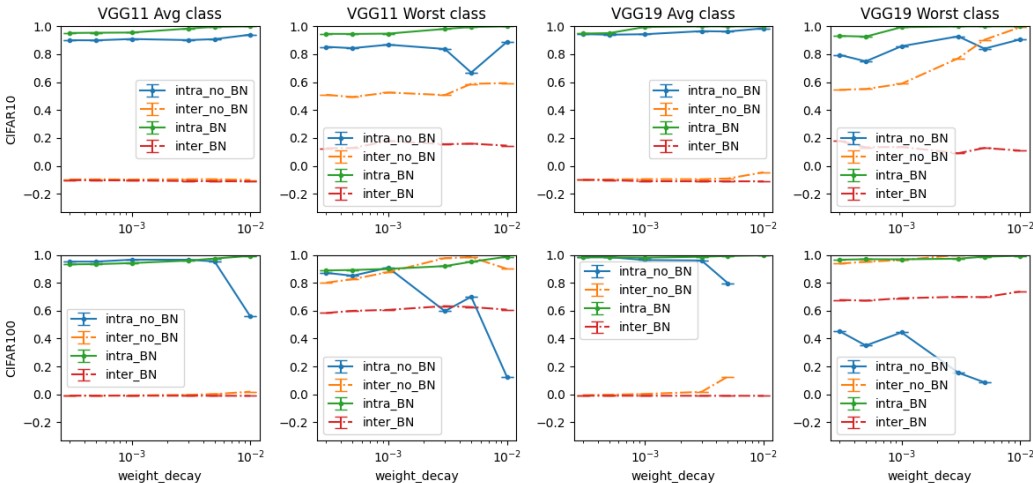

Figure 3: Intra-class and Inter-class Cosine Similarity for VGG11 and VGG19 under Different WD and BN. Higher intra-class and lower inter-class cosine similarity indicate a higher degree of $\mathcal{NC}$. Both the average measures over all classes and the worst class are presented. We note that the relationship between inter-class cosine similarity and weight decay is less pronounced in VGG models and does not increase or decrease significantly.

while the degree of $\mathcal{NC}$ is unstable or decreases with the increase of weight decay in non-BN models. Such a phenomenon is also more pronounced in simpler neural networks and easier classification tasks than in realistic classification tasks.

## 4    LIMITATIONS AND FUTURE WORK

Our theoretical exploration into deep neural network phenomena, specifically $\mathcal{NC}$, has its limitations and offers various avenues for further work. Based on our work, we have identified several directions for future efforts:

- Our work, like previous studies employing the layer-peeled model, primarily focuses on the last-layer features and posits that BN and weight decay are only applied to the penultimate layer. However, $\mathcal{NC}$ has been empirically observed in deeper network layers (Ben-Shaul & Dekel (2022); Galanti et al. (2022a)) and shown to be optimal for regularized MSE loss in deeper unconstrained features models (Tirer & Bruna (2022); Súkeník et al. (2023)). An insightful future direction would involve investigating how the proximity bounds to $\mathcal{NC}$ can be generalized to deeper layers of neural networks and understanding how these theoretical guarantees evolve with network depth.
- The theoretical model we have developed is idealized, omitting several intricate details inherent to practical neural networks. These include bias in linear layers and BN layers, and the sequence of BN and activation layers. Consequently, a worthwhile avenue for future research would be to refine the $\mathcal{NC}$ proximity bounds to accommodate more realistic network settings.

**Reproducibility Statement**    The proofs for all the theorems, lemmas, and propositions are included in the appendix. The assumptions for the theorems are clearly stated and can be fully interpreted with notations in Sections 2.1 and 2.2. For experimental results, all codes for experiments, raw data, and data visualization are included in the supplemental materials.

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

# A  PROOFS

## A.1  PROOF OF PROPOSITION 2.1

**Proposition 2.1.** *Let $\{\mathbf{h}_i\}_{i=1}^N$ be a set of feature vectors immediately after Batch Normalization with variance vector $\boldsymbol{\gamma}$ and bias term $\boldsymbol{\beta} = 0$ (i.e. $\mathbf{h}_i = BN(\mathbf{x}_i)$ for some $\{\mathbf{x}_i\}_{i=1}^N$). Then*

$$\sqrt{\frac{1}{N}\sum_{i=1}^N \|\mathbf{h}_i\|_2^2} = \|\boldsymbol{\gamma}\|_2$$

*Proof.* Let $\boldsymbol{\gamma}$ be the variance vector for the Batch Normalization layer, and consider a single batch $\{\mathbf{x}_i\}_{i=1}^B$ be a batch of $B$ vectors, and

$$h_i^{(k)} = \frac{x_i^{(k)} - \tilde{x}^{(k)}}{\sigma^{(k)}} \times \gamma^{(k)}$$

for all $B$. By the linearity of mean and standard deviation, $\hat{x}_i^{(k)} = \frac{x_i^{(k)} - \tilde{x}^{(k)}}{\sigma_{\mathbf{x}}^{(k)}}$ must have mean 0 and standard deviation 1. As a result, $\sum_{i=1}^B \hat{x}_i^{(k)} = 0$ and $\frac{1}{B}\sum_{i=1}^B (\hat{x}_i^{(k)})^2 = 1$. Therefore,

$$\sum_{i=1}^B (h_i^{(k)})^2 = \sum_{i=1}^B \gamma^{(k)}(\hat{x}_i^{(k)})^2 = B(\gamma^{(k)})^2$$

, and

$$\sum_{i=1}^B \|\mathbf{h}_i\|^2 = \sum_{k=1}^d \sum_{i=1}^B (h_i^{(k)})^2 = \sum_{k=1}^d \sum_{i=1}^B \gamma^{(k)}(\hat{x}_i^{(k)})^2 = \sum_{k=1}^d B(\gamma^{(k)})^2 = B\|\boldsymbol{\gamma}\|^2$$

.

Now, Consider a set of $N$ vectors divided into $m$ batches of size $\{B_j\}_{j=1}^m$. (This accounts for the fact that during training, the last mini-batch may have a different size than the other mini-batches if the number of training data is not a multiple of $B$). Then,

$$\sum_{i=1}^N \|\mathbf{h}_i\|^2 = \sum_{j=1}^m \sum_{i=1}^{B_j} \|\mathbf{h}_{j,i}\|^2 = \sum_{j=1}^m B_j\|\boldsymbol{\gamma}\|^2 = N\|\boldsymbol{\gamma}\|^2$$

Therefore, $\sqrt{\frac{1}{N}\sum_{i=1}^N \|\mathbf{h}_i\|^2} = \|\boldsymbol{\gamma}\|$ □

## A.2  PROOF OF LEMMA 2.1

**Lemma 2.1.** *Let $\{x_i\}_{i=1}^N \subset \mathcal{I}$ be a set of $N$ real numbers, let $\tilde{x} = \frac{1}{N}\sum_{i=1}^N x_i$ be the mean over all $x_i$ and $f$ be a function that is $m$-strongly-convex on $\mathcal{I}$. If*

$$\frac{1}{N}\sum_{i=1}^N f(x_i) \le f(\tilde{x}) + \epsilon$$

*Then for any subset of samples $S \subseteq [N]$, let $\delta = \frac{|S|}{N}$, there is*

$$\tilde{x} + \sqrt{\frac{2\epsilon(1-\delta)}{m\delta}} \ge \frac{1}{|S|}\sum_{i\in S} x_i \ge \tilde{x} - \sqrt{\frac{2\epsilon(1-\delta)}{m\delta}}$$

*Proof.* For the proof, we use a result from Merentes & Nikodem (2010) which bounds the Jensen inequality gap using the variance of the variables for strongly convex functions:

**Lemma A.1** (Theorem 4 from Merentes & Nikodem (2010))**.** *If $f : I \to \mathbb{R}$ is strongly convex with modulus $c$, then*

$$f\left(\sum_{i=1}^n t_i x_i\right) \leq \sum_{i=1}^n t_i f(x_i) - c \sum_{i=1}^n t_i (x_i - \bar{x})^2$$

*for all $x_1, \ldots, x_n \in I$, $t_1, \ldots, t_n > 0$ with $t_1 + \cdots + t_n = 1$ and $\bar{x} = t_1 x_1 + \cdots + t_n x_n$*

In the original definition of the authors, a strongly convex function with modulus $c$ is equivalent to a $2c$-strongly-convex function. We can apply $t_i = \frac{1}{N}$ for all $i$ and substitute the definition for strong convexity measure to obtain the following corollary:

**Corollary A.1.** *If $f : I \to \mathbb{R}$ is $m$-strongly-convex on $\mathcal{I}$, and*

$$\frac{1}{N}\sum_{i=1}^N f(x_i) = f\left(\frac{1}{N}\sum_{i=1}^N x_i\right) + \epsilon$$

*for $x_1, \ldots, x_N \in \mathcal{I}$, then $\frac{1}{N}\sum_i (x_i - \bar{x})^2 \leq \frac{2\epsilon}{m}$*

From A.1, we know that $\frac{1}{N}\sum_{i=1}^n (x_i - \tilde{x})^2 \leq \frac{2\epsilon}{m}$. Let $D = \sum_{i \in S}(x_i - \tilde{x})$, by the convexity of $x^2$, there is

$$\sum_{i=1}^n (x_i - \tilde{x})^2 = \sum_{i \in S}(x_i - \tilde{x})^2 + \sum_{i \notin S}(x_i - \tilde{x})^2$$

$$\geq |S|\left(\frac{1}{|S|}\sum_{i \in S}(x_i - \tilde{x})\right)^2 + (N - |S|)\left(\frac{1}{N-|S|}\sum_{i \notin S}(x_i - \tilde{x})\right)^2$$

$$= \frac{1}{S}\left(\sum_{i \in S}(x_i - \tilde{x})\right)^2 + \frac{1}{N-|S|}\left(\sum_{i \notin S}(x_i - \tilde{x})\right)^2$$

$$= \frac{1}{S}D^2 + \frac{1}{N-|S|}(-D)^2$$

$$= \frac{D^2}{N}\left(\frac{1}{\delta} + \frac{1}{1-\delta}\right)$$

$$= \frac{D^2}{N}\left(\frac{1}{\delta(1-\delta)}\right)$$

Therefore $\frac{D^2}{N}\left(\frac{1}{\delta(1-\delta)}\right) \leq \frac{2\epsilon N}{m}$, and $|D| \leq \sqrt{\frac{2\epsilon\delta(1-\delta)N^2}{\lambda}}$. Using $\frac{1}{|S|}\sum_{i \in S} x_i = \frac{1}{|S|}(|S|\tilde{x} + D)$ and $|S| = \delta N$ completes the proof. $\qquad\square$

### A.3 Proof of Theorem 2.1

Note that compared to the theorem in the main text, this theorem includes the non-asymptotic representation and the corresponding result for NC3.

**Theorem 2.1.** *For any neural network classifier without bias terms trained on dataset with the number of classes $C \geq 3$ and samples per class $N \geq 1$, under the following assumptions:*

1. *The quadratic average of the feature norms $\sqrt{\frac{1}{CN}\sum_{c=1}^C \sum_{i=1}^N \|\mathbf{h}_{c,i}\|^2} \leq \alpha$*

2. *The Frobenius norm of the last-layer weight $\|\mathbf{W}\|_F \leq \sqrt{C}\beta$*

3. *The average cross-entropy loss over all samples $\mathcal{L} \leq m + \epsilon$ for small $\epsilon$*

*where $m = \log(1 + (C-1)\exp(-\frac{C}{C-1}\alpha\beta))$ is the minimum achievable loss for any set of weight and feature vectors satisfying the norm constraints, then for at least $1 - \delta$ fraction of all classes , with $\frac{\epsilon}{\delta} \ll 1$, for small constant $\kappa > 0$ there is*

$$intra_c \geq 1 - \frac{C-1}{C\alpha\beta}\sqrt{\frac{128\epsilon(1-\delta)\exp(\kappa C\alpha\beta)}{\delta}} = 1 - O\left(\frac{e^{O(C\alpha\beta)}}{\alpha\beta}\sqrt{\frac{\epsilon}{\delta}}\right),$$

*and also for a cosine similarity representation of NC3 in Papyan et al. (2020):*

$$\cos_{\angle}(\dot{\mathbf{w}}_c, \tilde{\mathbf{h}}_c) \geq 1 - 2\sqrt{\frac{2\epsilon(1-\delta)e^{\kappa C\alpha\beta}}{\delta}} = 1 - O(e^{O(C\alpha\beta)}\sqrt{\frac{\epsilon}{\delta}}),$$

*and for at least $1 - \delta$ fraction of all pairs of classes $c, c'$, with $\frac{\epsilon}{\delta} \ll 1$, there is*

$$inter_{c,c'} \leq -\frac{1}{C-1} + \frac{C}{C-1}\frac{\exp(\kappa C\alpha\beta)}{\alpha\beta}\sqrt{\frac{2\epsilon}{\delta}} + 4(\frac{2\exp(\kappa C\alpha\beta)}{\alpha\beta}\sqrt{\frac{2\epsilon}{\delta}})^{1/3} + \sqrt{\frac{\exp(\kappa C\alpha\beta)}{\alpha\beta}\sqrt{\frac{2\epsilon}{\delta}}}$$

$$= -\frac{1}{C-1} + O(\frac{e^{O(C\alpha\beta)}}{\alpha\beta}(\frac{\epsilon}{\delta})^{1/6})$$

We first present several lemmas that facilitate the proof technique used in the main proof. The first two lemmas demonstrate that if a set of variables achieves roughly equal value on the LHS and RHS of Jensen's inequality for a strongly convex function, then the mean of every subset cannot deviate too far from the global mean.

Our first lemma states that, For $\lambda$-strongly-convex-function $f$ and a set of numbers $\{x_i\}_{i=1}^N$, if Jensen's inequality has its gap bounded by $\epsilon$, then the mean of any subset that includes $\delta$ fraction of all samples can not deviate from global mean of all samples by more than $\sqrt{\frac{2\epsilon(1-\delta)}{\lambda\delta}}$:

Our second lemma states a similar result specific to the function $e^x$ and only provides the upper bound. Note that, within any predefined range $[a, b]$, $\exp(x)$ can only be guaranteed to be $e^a$ strongly convex, which may be bad if the lower bound $a$ is small or does not exist. Our further result in the following lemma shows that we can provide a better upper bound of the subset mean for the exponential function that is dependent on $\exp(\tilde{x})$ and does not require other prior knowledge of the range of $x_i$:

**Lemma A.2.** *Let $\{x_i\}_{i=1}^N \subset \mathbb{R}$ be any set of $N$ real numbers, let $\tilde{x} = \frac{1}{N}\sum_{i=1}^N x_i$ be the mean over all $x_i$. If*

$$\frac{1}{N}\sum_{i=1}^N \exp(x_i) \leq \exp(\tilde{x}) + \epsilon$$

*then for any subset $S \subseteq [N]$, let $\delta = \frac{|S|}{N}$, the there is*

$$\frac{1}{|S|}\sum_{i\in S} x_i \leq \tilde{x} + \sqrt{\frac{2\epsilon}{\delta\exp(\tilde{x})}}.$$

*Proof.* Let $D = \sum_{i\in S}(x_i - \tilde{x})$. Note that if $D < 0$ then the upper bound is obviously satisfied since the subset mean will be smaller than the global mean. Therefore, we only consider the case when

$D > 0$

$$\sum_{i=1}^{N} \exp(x_i) = \sum_{i \in S} \exp(x_i) + \sum_{i \notin S} \exp(x_i)$$

$$\geq |S| \exp(\frac{1}{|S|} \sum_{i \in S} x_i) + (N - |S|) \exp(\frac{1}{N - |S|} \sum_{i \notin S} x_i)$$

$$\geq |S| \exp(\tilde{x} + \frac{D}{|S|}) + (N - |S|) \exp(\tilde{x} - \frac{D}{N - |S|})$$

$$\geq |S| \exp(\tilde{x})(1 + \frac{D}{|S|} + \frac{D^2}{2|S|^2}) + (N - |S|) \exp(\tilde{x})(1 - \frac{D}{N - |S|})$$

$$= (N + \frac{D^2}{2|S|}) \exp(\tilde{x})$$

$$N \exp(\tilde{x}) + N\epsilon \geq (N + \frac{D^2}{2|S|}) \exp(\tilde{x})$$

$$D^2 \leq \frac{2|S|N\epsilon}{\exp(\tilde{x})}$$

$$D \leq N \sqrt{\frac{2\delta\epsilon}{\exp(\tilde{x})}}$$

Using $\frac{1}{|S|} \sum_{i \in S} x_i = \frac{1}{|S|}(|S|\tilde{x} + D)$ and $|S| = \delta N$ completes the proof. $\qquad\square$

Directly approaching the average intra-class and inter-class cosine similarity of vector set(s) is a relatively difficult task. Our following lemma shows that the inter-class and inter-class cosine similarities can be computed as the norm and dot product of the vectors $\tilde{\mathbf{h}}_c$, respectively, where $\tilde{\mathbf{h}}_c$ is the mean *normalized* vector among all vectors in a class.

**Lemma A.3.** *Let $c, c'$ be 2 classes, each containing $N$ feature vectors $\mathbf{h}_{c,i} \in \mathbb{R}^d$. Define the average intra-class cosine similarity of picking two vectors from the same class $c$ as*

$$intra_c = \frac{1}{N^2} \sum_{i=1}^{N} \sum_{j=1}^{N} \cos_\angle(\mathbf{h}_{c,i}, \mathbf{h}_{c,j})$$

*and the intra-class cosine similarity between two classes $c, c'$ is defined as the average cosine similarity of picking one feature vector of class $c$ and another from class $c'$ as*

$$inter_c = \frac{1}{N^2} \sum_{i=1}^{N} \sum_{j=1}^{N} \cos_\angle(\mathbf{h}_{c,i}, \mathbf{h}_{c',j})$$

*Let $\tilde{\mathbf{h}}_c = \frac{1}{N} \sum_{i=1}^{N} \frac{\mathbf{h}_{c,i}}{\|\mathbf{h}_{c,i}\|}$. Then $intra_c = \|\tilde{\mathbf{h}}_c\|^2$ and $inter_{c,c'} = \tilde{\mathbf{h}}_c \cdot \tilde{\mathbf{h}}_{c'}$*

*Proof.* For the intra-class cosine similarity,

$$intra_c = \frac{1}{N^2} \sum_{i=1}^{N} \sum_{j=1}^{N} \bar{\mathbf{h}}_{c,i} \cdot \bar{\mathbf{h}}_{c,j}$$

$$= \frac{1}{N^2} \sum_{i=1}^{N} \sum_{j=1}^{N} \frac{\mathbf{h}_{c,i}}{\|\mathbf{h}_{c,i}\|} \cdot \frac{\mathbf{h}_{c,j}}{\|\mathbf{h}_{c,j}\|}$$

$$= \frac{1}{N^2} \sum_{i=1}^{N} \sum_{j=1}^{N} \frac{\mathbf{h}_{c,i} \cdot \mathbf{h}_{c,j}}{\|\mathbf{h}_{c,i}\|\|\mathbf{h}_{c,j}\|}$$

$$= \left( \frac{1}{N} \sum_{i=1}^{N} \frac{\mathbf{h}_{c,i}}{\|\mathbf{h}_{c,i}\|} \right) \cdot \left( \frac{1}{N} \sum_{j=1}^{N} \frac{\mathbf{h}_{c,j}}{\|\mathbf{h}_{c,j}\|} \right)$$

$$= \|\tilde{\bar{\mathbf{h}}}_c\|^2$$

and for the inter-class cosine similarity,

$$inter_{c,c'} = \frac{1}{N^2} \sum_{i=1}^{N} \sum_{j=1}^{N} \bar{\mathbf{h}}_{c,i} \cdot \bar{\mathbf{h}}_{c',j}$$

$$= \frac{1}{N^2} \sum_{i=1}^{N} \sum_{j=1}^{N} \frac{\mathbf{h}_{c,i}}{\|\mathbf{h}_{c,i}\|} \cdot \frac{\mathbf{h}_{c',j}}{\|\mathbf{h}_{c',j}\|}$$

$$= \frac{1}{N^2} \sum_{i=1}^{N} \sum_{j=1}^{N} \frac{\mathbf{h}_{c,i} \cdot \mathbf{h}_{c',j}}{\|\mathbf{h}_{c,i}\|\|\mathbf{h}_{c',j}\|}$$

$$= \left( \frac{1}{N} \sum_{i=1}^{N} \frac{\mathbf{h}_{c,i}}{\|\mathbf{h}_{c,i}\|} \right) \cdot \left( \frac{1}{N} \sum_{j=1}^{N} \frac{\mathbf{h}_{c',j}}{\|\mathbf{h}_{c',j}\|} \right)$$

$$= \tilde{\bar{\mathbf{h}}}_c \cdot \tilde{\bar{\mathbf{h}}}_{c'}$$

$\square$

We prove the intra-class cosine similarity by first showing that the norm of the mean (un-normalized) class-feature vector for a class is near the quadratic average of feature means (i.e., $\|\tilde{\mathbf{h}}_c\| = \|\frac{1}{N} \sum_{i=1}^{N} \mathbf{h}_{c,i}\| \approx \sqrt{\frac{1}{N} \sum_{i=1}^{N} \|\mathbf{h}_{c,i}\|^2}$). However, to show intra-class cosine similarity, we need instead a bound on $\|\tilde{\bar{\mathbf{h}}}_c\| = \|\frac{1}{N} \sum_{i=1}^{N} \bar{\mathbf{h}}_{c,i}\|$. The following lemma provides a conversion between these requirements:

**Lemma A.4.** *Suppose* $\mathbf{u} \in \mathbb{R}^d$ *and* $\|\mathbf{u}\| \leq \beta$. *Let* $\{\mathbf{v}_i\}_{i=1}^{N} \subset \mathbb{R}^d$ *such that* $\frac{1}{N} \|\mathbf{v}_i\|^2 \leq \alpha^2$. *If*

$$\frac{1}{N} \sum_{i=1}^{N} \langle \mathbf{u}, \mathbf{v}_i \rangle \geq c,$$

*for* $\frac{\alpha\beta}{\sqrt{2}} \leq c \leq \alpha\beta$ *and let* $\bar{\mathbf{v}} = \frac{\mathbf{v}}{\|\mathbf{v}\|}$ *then*

$$\tilde{\bar{\mathbf{v}}} = \|\frac{1}{N} \sum_{i=1}^{N} \bar{\mathbf{v}}\| \geq 2(\frac{c}{\alpha\beta})^2 - 1.$$

*Proof.* Divide into 2 cases: the set of indices

$$pos = \{i \in [N] | \langle \mathbf{u}, \mathbf{v}_i \rangle > 0\}$$

and

$$neg = \{i \in [N] | \langle \mathbf{u}, \mathbf{v}_i \rangle < 0\}$$

Let $M = |pos|$, then note that

$$\sum_{i \in pos} \langle \mathbf{u}, \mathbf{v}_i \rangle \geq Nc$$

$$\sum_{i \in pos} \langle \mathbf{u}, \mathbf{v}_i \rangle \leq \|\mathbf{u}\| \sum_{i \in pos} \|\mathbf{v}_i\|$$

$$\leq \beta \sqrt{M \sum_{i \in pos} \|\mathbf{v}_i\|^2} \qquad\qquad E[X^2] \geq E[X]^2$$

$$\leq \beta \sqrt{MN\alpha^2}$$

$$= \alpha\beta\sqrt{MN}$$

Therefore

$$N \geq M \geq N(\frac{c}{\alpha\beta})^2$$

First, consider $\sum_{i \in pos} \langle \mathbf{u}, \bar{\mathbf{v}}_i \rangle$. Note that $\sum_{i \in pos} \|\mathbf{v}_i\|^2 \leq N\alpha^2$ and $\sum_{i \in pos} \langle \mathbf{u}, \mathbf{v}_i \rangle \geq Nc$. We will use the following proposition that can be easily shown through Lagrange multipliers: Given $\{a_i\}_{i=1}^N$ and $\{b_i\}_{i=1}^N$ such that $a_i \geq 0$ and $b_i > 0$ for all $i$, if $\sum_{i=1}^n a_i = A$ and $\sum_{i=1}^n b_i^2 \leq B$, then $\sum_{i=1}^N \frac{a_i}{b_i} \geq \frac{A\sqrt{N}}{\sqrt{B}}$

Therefore

$$\sum_{i \in pos} \langle \mathbf{u}, \bar{\mathbf{v}}_i \rangle = \sum_{i \in pos} \frac{\langle \mathbf{u}, \mathbf{v}_i \rangle}{\|\mathbf{v}_i\|}$$

$$\geq \frac{c}{\alpha} \cdot \sqrt{MN} \qquad\qquad \text{Proposition}$$

$$\geq \frac{c}{\alpha} \cdot (N\frac{c}{\alpha\beta})$$

$$= N\beta(\frac{c}{\alpha\beta})^2$$

On the other hand, for $neg$, since $\langle \mathbf{u}, \bar{\mathbf{v}}_i \rangle \geq -\|u\| \geq -\beta$, we get

$$\sum_{i \in pos} \langle \mathbf{u}, \bar{\mathbf{v}}_i \rangle \geq \sum_{i \in neg} -\beta$$

$$= -\beta(N - M)$$

$$\geq -\beta N(1 - (\frac{c}{\alpha\beta})^2) = N\beta((\frac{c}{\alpha\beta})^2 - 1)$$

Therefore

$$\|\mathbf{u}\| \|\frac{1}{N} \sum_{i=1}^N \bar{\mathbf{v}}_i\| \geq \frac{1}{N} \sum_{i=1}^N \langle \mathbf{u}, \bar{\mathbf{v}}_i \rangle \geq \frac{1}{N}(\sum_{i \in pos} \langle \mathbf{u}, \bar{\mathbf{v}}_i \rangle + \sum_{i \in neg} \langle \mathbf{u}, \bar{\mathbf{v}}_i \rangle)$$

$$\geq \frac{1}{N}(N\beta((\frac{c}{\alpha\beta})^2 - 1) + N\beta(\frac{c}{\alpha\beta})^2)$$

$$= \beta(2(\frac{c}{\alpha\beta})^2 - 1)$$

$$\|\frac{1}{N} \sum_{i=1}^N \bar{\mathbf{v}}\|^2 \geq 2(\frac{c}{\alpha\beta})^2 - 1$$

$$\square$$

To make this lemma generalize to other proofs in future work, we provide the generalized corollary of the above lemma by setting $\mathbf{u}$ to be the normalized mean vector of $\mathbf{v}$:

**Corollary A.2.** *Let* $\{\mathbf{v}_i\}_{i=1}^N \subset \mathbb{R}^d$ *such that* $\frac{1}{N}\|\mathbf{v}_i\|^2 \le \alpha^2$. *If*

$$\|\frac{1}{N}\sum_{i=1}^N \mathbf{v}_i\| \ge c,$$

*for* $\frac{\alpha}{\sqrt{2}} \le c \le \alpha$ *and let* $\bar{\mathbf{v}} = \frac{\mathbf{v}}{\|\mathbf{v}\|}$ *then*

$$\tilde{\bar{\mathbf{v}}} = \|\frac{1}{N}\sum_{i=1}^N \bar{\mathbf{v}}\| \ge 2(\frac{c}{\alpha})^2 - 1.$$

Similarly, for inter-class cosine similarity, we have the following lemma:

**Lemma A.5.** *Let* $\mathbf{w} \in \mathbb{R}^d$, $\{\mathbf{h}_i\}_{i=1}^N \subset \mathbb{R}^d$. *Let* $\tilde{\mathbf{h}} = \frac{1}{N}\sum_{i=1}^N \mathbf{h}_i$ *and* $\tilde{\bar{\mathbf{h}}} = \frac{1}{N}\sum_{i=1}^N \frac{\mathbf{h}_i}{\|\mathbf{h}_i\|}$. *If the following condition is satisfied:*

$$\frac{1}{N}\mathbf{w}\cdot\sum_{i=1}^N \mathbf{h}_i \le c \qquad\qquad for\ c < 0$$

$$\|\mathbf{w}\| \le \beta$$

$$\frac{1}{N}\sum_{i=1}^n \|\mathbf{h}_i\|^2 \le \alpha^2$$

$$\exists \mathbf{w}' \in \mathbb{R}^d, \|\mathbf{w}'\| \le \beta, \frac{1}{N}\mathbf{w}'\sum_{i=1}^N \mathbf{h}_i \ge \alpha\beta - \epsilon'$$

$$\epsilon' \ll \alpha\beta$$

*Then* $\cos_\angle(\mathbf{w}, \tilde{\bar{\mathbf{h}}}) \le -\frac{c}{\alpha\beta} + 4(\frac{\epsilon'}{\alpha\beta})^{1/3}$

*Proof.* For $\mathbf{w} \in \mathbb{R}^d$, $\{\mathbf{h}_i\}_{i=1}^N \subset \mathbb{R}^d$

Let $a_i = \frac{1}{N}\mathbf{w}\mathbf{h}_i$, $b_i = \|\mathbf{h}_i\|$, $\epsilon = \frac{\epsilon'}{\beta}$, then the constraints of the above problem relaxed as follows:

$$\max \sum_{i=1}^N \frac{a_i}{b_i}$$

$$s.t. \sum_{i=1}^N a_i \le c$$

$$\frac{1}{N}\sum_{i=1}^N b_i^2 = \alpha^2$$

$$\frac{1}{N}\sum_{i=1}^N b_i \ge \alpha - \frac{\epsilon}{\beta}$$

$$\forall i, |\frac{a_i}{b_i}| \le \beta.$$

First, consider the case when $\frac{1}{N}\sum_{i=1}^N b_i \ge \alpha - \epsilon$ Consider a random variable $B$ that uniformly picks a value from $\{b_i\}_{i=1}^N$. Then $\mathbb{E}[B] \ge \alpha - \frac{\epsilon}{\beta}$, $\mathbb{E}[B^2] = \alpha^2$, and therefore $\sigma_B = \sqrt{\mathbb{E}[B^2] - \mathbb{E}[B]^2} \le \sqrt{2\alpha\epsilon}$. According to Chebyshev's inequality

$$P(|B - (\alpha - \epsilon)| \ge k\sqrt{2\alpha\epsilon}) \le \frac{1}{k^2}.$$

Note that for positive $a_i$, smaller $b_i$ means larger $\frac{a_i}{b_i}$ and for negative $a_i$, higher $b_i$ means larger $\frac{a_i}{b_i}$. Suppose that $\epsilon$ is sufficiently small such that $\epsilon \ll \sqrt{\epsilon}$. Therefore, an upper bound for $\frac{a_i}{b_i}$ when $a_i > 0$ is

$$\frac{a_i}{b_i} \le \begin{cases} \frac{a_i}{\alpha - k\sqrt{2\alpha\epsilon}} & b_i \ge \alpha - k\sqrt{2\alpha\epsilon} \\ \beta & b_i < \alpha - k\sqrt{2\alpha\epsilon} \end{cases},$$

and an upper bound for $a_i < 0$ would is

$$\frac{a_i}{b_i} \leq \begin{cases} \frac{a_i}{\alpha + k\sqrt{2\alpha\epsilon}} & b_i \leq \alpha + k\sqrt{2\alpha\epsilon} \\ 0 & b_i > \alpha + k\sqrt{2\alpha\epsilon} \end{cases}.$$

Suppose that $k\sqrt{\frac{2\epsilon}{\alpha}}$ is less than $\frac{1}{2}$, then

$$\frac{a_i}{\alpha - k\sqrt{2\alpha\epsilon}} = \frac{a_i}{\alpha} \cdot \frac{1}{1 - k\sqrt{\frac{2\epsilon}{\alpha}}} < \frac{a_i}{\alpha} \cdot (1 + 2k\sqrt{\frac{2\epsilon}{\alpha}}) = \frac{a_i}{\alpha} + |\frac{a_i}{\alpha}| \cdot 2k\sqrt{\frac{2\epsilon}{\alpha}}$$

when $a_i > 0$, and similarly

$$\frac{a_i}{\alpha + k\sqrt{2\alpha\epsilon}} = \frac{a_i}{\alpha} \cdot \frac{1}{1 + k\sqrt{\frac{2\epsilon}{\alpha}}} < \frac{a_i}{\alpha} \cdot (1 - 2k\sqrt{\frac{2\epsilon}{\alpha}}) = \frac{a_i}{\alpha} + |\frac{a_i}{\alpha}| \cdot 2k\sqrt{\frac{2\epsilon}{\alpha}}$$

when $a_i < 0$. Note that

$$\sum_{i=1}^{N} |\frac{a_i}{\alpha}| \cdot 2k\sqrt{\frac{2\epsilon}{\alpha}} \leq \sum_{i=1}^{N} \frac{\beta}{N} \cdot 2k\sqrt{\frac{2\epsilon}{\alpha}} = 2k\beta\sqrt{\frac{2\epsilon}{\alpha}}$$

Therefore, an upper bound on the total sum would be:

$$\frac{c}{\alpha} + 2k\beta\sqrt{\frac{2\epsilon}{\alpha}} + \frac{\beta}{k^2}$$

Set $k = (\sqrt{\frac{8\epsilon}{\alpha}})^{-\frac{1}{3}}$ to get:

$$\frac{c}{\alpha} + 2\beta(\sqrt{\frac{8\epsilon}{\alpha}})^{\frac{2}{3}} = \frac{c}{\alpha} + 4\beta(\frac{\epsilon}{\alpha})^{\frac{1}{3}}$$

Now, we substitute $\epsilon = \frac{\epsilon'}{\beta}$ we get: $\mathbf{w} \cdot \tilde{\mathbf{h}} \leq \frac{c}{\alpha} + 4\beta(\frac{\epsilon'}{\alpha\beta})^{1/3}$ Since $|\mathbf{w}| \leq \beta$ and $|\tilde{\mathbf{h}}| \leq 1$, we get that

$$\cos_{\angle}(\mathbf{w}, \tilde{\mathbf{h}}) \leq \frac{c}{\alpha\beta} + 4(\frac{\epsilon'}{\alpha\beta})^{1/3}$$

$\square$

Now we proceed to the main proof: First, consider the minimum achievable average loss for a single class $c$:

$$\frac{1}{N} \sum_{i=1}^{N} L_{c,i} = \frac{1}{N} \sum_{i=1}^{N} \text{softmax}(\mathbf{W}\mathbf{h}_{c,i})_c \tag{1}$$

$$\geq \text{softmax}(\frac{1}{N} \sum_{i=1}^{N} \mathbf{W}\mathbf{h}_{c,i})_c \tag{2}$$

$$= \log\left(1 + \sum_{c' \neq c} \exp(\frac{1}{N} \sum_{i=1}^{N} (\mathbf{w}_{c'} - \mathbf{w}_c)\mathbf{h}_{c,i})\right) \tag{3}$$

$$= \log\left(1 + \sum_{c' \neq c} \exp((\mathbf{w}_{c'} - \mathbf{w}_c)\tilde{\mathbf{h}}_c)\right) \tag{4}$$

$$\geq \log\left(1 + (C - 1)\exp(\frac{1}{(C-1)}(\sum_{c'=1}^{C} \mathbf{w}_{c'}\tilde{\mathbf{h}}_c - C\mathbf{w}_c\tilde{\mathbf{h}}_c))\right) \tag{5}$$

$$= \log\left(1 + (C - 1)\exp(\frac{1}{(C-1)}(\sum_{c'=1}^{C} \mathbf{w}_{c'} - C\mathbf{w}_c)\tilde{\mathbf{h}}_c)\right) \tag{6}$$

$$= \log\left(1 + (C - 1)\exp(\frac{C}{C-1}(\tilde{\mathbf{w}} - \mathbf{w}_c)\tilde{\mathbf{h}}_c)\right) \tag{7}$$

$$= \log\left(1 + (C - 1)\exp(-\frac{C}{C-1}\dot{\mathbf{w}}_c\tilde{\mathbf{h}}_c)\right) \tag{8}$$

Let $\overrightarrow{\mathbf{w}} = [\mathbf{w}_1 - \tilde{\mathbf{w}}, \mathbf{w}_2 - \tilde{\mathbf{w}}, \ldots, \mathbf{w}_C - \tilde{\mathbf{w}}] = [\dot{\mathbf{w}}_1, \dot{\mathbf{w}}_2, \ldots, \dot{\mathbf{w}}_C]$, and $\overrightarrow{\mathbf{h}} = [\tilde{\mathbf{h}}_1, \tilde{\mathbf{h}}_2, \ldots, \tilde{\mathbf{h}}_c] \in \mathbf{R}^{Cd}$.
Note that

$$\|\overrightarrow{\mathbf{w}}\|^2 = \sum_{c=1}^{C} \|\mathbf{w}_c - \tilde{\mathbf{w}}\|^2 = \sum_{c=1}^{C} \left( \|\mathbf{w}_c\|^2 - 2\mathbf{w}_c\tilde{\mathbf{w}} + \|\tilde{\mathbf{w}}\|^2 \right)$$

$$= \sum_{c=1}^{C} \|\mathbf{w}_c\|^2 - C\|\tilde{\mathbf{w}}\|^2 \leq \sum_{c=1}^{C} \|\mathbf{w}_c\|^2 = \|\mathbf{W}\|_F^2 \leq C\beta^2$$

and also

$$\|\overrightarrow{\mathbf{h}}\|^2 = \sum_{c=1}^{C} \|\tilde{\mathbf{h}}_c\|^2 = \sum_{c=1}^{C} \|\frac{1}{N} \sum_{i=1}^{N} \mathbf{h}_{c,i}\|^2 \leq \sum_{c=1}^{C} \left( \frac{1}{N} \sum_{i=1}^{N} \|\mathbf{h}_{c,i}\| \right)^2$$

$$\leq \frac{1}{N} \sum_{c=1}^{C} \sum_{i=1}^{N} \|\mathbf{h}_{c,i}\|^2 = C\alpha^2$$

The first inequality uses the triangle inequality and the second uses $\mathbb{E}[X^2] \geq \mathbb{E}[X]^2$ Now consider the total average loss over all classes:

$$\mathcal{L} = \frac{1}{CN} \sum_{c=1}^{C} \sum_{i=1}^{N} L_{c,i}$$

$$\geq \frac{1}{C} \sum_{c=1}^{C} \log \left( 1 + (C-1) \exp(\frac{C}{C-1}(\tilde{\mathbf{w}} - \mathbf{w}_c)\tilde{\mathbf{h}}_c) \right)$$

$$\geq \log \left( 1 + (C-1) \exp(\frac{C}{C-1} \cdot \frac{1}{C} \sum_{c=1}^{C} (\tilde{\mathbf{w}} - \mathbf{w}_c)\tilde{\mathbf{h}}_c) \right) \qquad \text{Jensen's}$$

$$\geq \log \left( 1 + (C-1) \exp(-\frac{1}{C-1} \overrightarrow{\mathbf{w}} \cdot \overrightarrow{\mathbf{h}}) \right)$$

$$\geq \log \left( 1 + (C-1) \exp(-\frac{C}{C-1} \alpha\beta) \right)$$

$$= m,$$

showing that $m$ is indeed the minimum achievable average loss among all samples.
Now we instead consider when the final average loss is near-optimal of value $m + \epsilon$ with $\epsilon \ll 1$. We use a new $\epsilon$ to represent the gap introduced by each inequality in the above proof. Additionally, since

the average loss is near-optimal, there must be $\dot{\mathbf{w}}_c\tilde{\mathbf{h}}_c \geq 0$ for any sufficiently small $\epsilon$:

$$\frac{1}{N}\sum_{i=1}^{N}L_{c,i} = \frac{1}{N}\sum_{i=1}^{N}\text{softmax}(\mathbf{W}\mathbf{h}_{c,i})_c \tag{9}$$

$$\geq \text{softmax}(\frac{1}{N}\sum_{i=1}^{N}\mathbf{W}\mathbf{h}_{c,i})_c \tag{10}$$

$$= \log\left(1 + \sum_{c'\neq c}\exp(\frac{1}{N}\sum_{i=1}^{N}\mathbf{w}_{c'}\mathbf{h}_{c,i} - \frac{1}{N}\sum_{i=1}^{N}\mathbf{w}_c\mathbf{h}_{c,i})\right) \tag{11}$$

$$= \log\left(1 + \sum_{c'\neq c}\exp(\frac{1}{N}\sum_{i=1}^{N}(\mathbf{w}_{c'} - \mathbf{w}_c)\mathbf{h}_{c,i})\right) \tag{12}$$

$$= \log\left(1 + \sum_{c'\neq c}\exp((\mathbf{w}_{c'} - \mathbf{w}_c)\tilde{\mathbf{h}}_c)\right) \tag{13}$$

$$= \log\left(1 + (C-1)\exp(\frac{1}{(C-1)}(\sum_{c'=1}^{C}\mathbf{w}_{c'}\tilde{\mathbf{h}}_c - C\mathbf{w}_c\tilde{\mathbf{h}}_c)) + \epsilon_1'\right) \tag{14}$$

$$= \log\left(1 + (C-1)\exp(\frac{1}{(C-1)}(\sum_{c'=1}^{C}\mathbf{w}_{c'} - C\mathbf{w}_c)\tilde{\mathbf{h}}_c) + \epsilon_1'\right) \tag{15}$$

$$= \log\left(1 + (C-1)\exp(\frac{C}{C-1}(\tilde{\mathbf{w}} - \mathbf{w}_c)\tilde{\mathbf{h}}_c) + \epsilon_1'\right) \tag{16}$$

$$\geq \log\left(1 + (C-1)\exp(-\frac{C}{C-1}\dot{\mathbf{w}}_c\tilde{\mathbf{h}}_c)\right) + \frac{\epsilon_1'}{1 + (C-1)\exp(-\frac{C}{C-1}\dot{\mathbf{w}}_c\tilde{\mathbf{h}}_c)} \tag{17}$$

$$\geq \log\left(1 + (C-1)\exp(-\frac{C}{C-1}\dot{\mathbf{w}}_c\tilde{\mathbf{h}}_c)\right) + \frac{\epsilon_1'}{C} \tag{18}$$

and also

$$\mathcal{L} = \frac{1}{CN}\sum_{c=1}^{C}\sum_{i=1}^{N}L_{c,i}$$

$$\geq \frac{1}{C}\sum_{c=1}^{C}\left(\log\left(1 + (C-1)\exp(-\frac{C}{C-1}\dot{\mathbf{w}}_c\tilde{\mathbf{h}}_c)\right) + \frac{\epsilon_{1,c}'}{C}\right)$$

$$= \log\left(1 + (C-1)\exp(-\frac{C}{C-1}\cdot\frac{1}{C}\sum_{c=1}^{C}\dot{\mathbf{w}}_c\tilde{\mathbf{h}}_c)\right) + \frac{1}{C}\sum_{c=1}^{C}\frac{\epsilon_{1,c}'}{C} + \epsilon_2' \qquad \text{Jensen's}$$

$$= \log\left(1 + (C-1)\exp(-\frac{1}{C-1}\overrightarrow{\mathbf{w}}\cdot\overrightarrow{\mathbf{h}})\right) + \frac{1}{C}\sum_{c=1}^{C}\frac{\epsilon_{1,c}'}{C} + \epsilon_2'$$

$$= \log\left(1 + (C-1)\exp(-\frac{C}{C-1}\alpha\beta + \epsilon_3')\right) + \frac{1}{C}\sum_{c=1}^{C}\frac{\epsilon_{1,c}'}{C} + \epsilon_2'$$

Consider $\log(1 + (C-1)\exp(-\frac{C\alpha\beta}{C-1} + \epsilon_3'))$: Let $\gamma' = (C-1)\exp(-\frac{C\alpha\beta}{C-1})$

$$
\begin{aligned}
\log(1 + (C-1)\exp(-\frac{C\alpha\beta}{C-1} + \epsilon_3')) &= \log(1 + (C-1)\exp(-\frac{C\alpha\beta}{C-1})\exp(\epsilon_3')) \\
&= \log(1 + (C-1)\exp(-\frac{C\alpha\beta}{C-1})\exp(\epsilon_3')) \\
&= \log(1 + \gamma'\exp(\epsilon_3')) \\
&\geq \log(1 + \gamma'(1 + \epsilon_3')) \\
&= \log(1 + \gamma' + \gamma'\epsilon_3') \\
&\approx \log(1 + \gamma') + \frac{\gamma'}{1+\gamma'}\epsilon_3'
\end{aligned}
$$

Thus using the fact that $1 + (C-1)\exp(-\frac{C\alpha\beta}{C-1}) \leq C$

$$
\mathcal{L} \geq \log(1 + (C-1)\exp(-\frac{C\alpha\beta}{C-1})) + \frac{1}{C}\sum_{c=1}^{C}\frac{\epsilon_{1,c}'}{C} + \epsilon_2' + \frac{\gamma'}{1+\gamma'}\epsilon_3'
$$

$$
\epsilon \geq \frac{1}{C}\sum_{c=1}^{C}\frac{\epsilon_{1,c}'}{C} + \epsilon_2' + \frac{\gamma'}{1+\gamma'}\epsilon_3'
$$

Note that while we do not know how $\epsilon$ is distributed among the different gaps, all the bounds involving $\epsilon_{1,c}', \epsilon_2', \epsilon_3'$ always hold in the worst case scenario subject to the constraint $\epsilon \geq \frac{1}{C}\sum_{c=1}^{C}\frac{\epsilon_{1,c}'}{C} + \epsilon_2' + \frac{\gamma'}{1+\gamma'}\epsilon_3'$. Note that $\|\tilde{\mathbf{h}}_c\| \leq \sum_{c'=1}^{C}\|\tilde{\mathbf{h}}_{c'}\| \leq \sqrt{C}\alpha$, and $\|\dot{\mathbf{w}}_c\| \leq \|W\|_F = \sqrt{C}\beta$ therefore $\dot{\mathbf{w}}_c\tilde{\mathbf{h}}_c \geq -C\alpha\beta$. We also know that

$$
\frac{1}{C}\sum_{c=1}^{C}\dot{\mathbf{w}}_c\tilde{\mathbf{h}}_c \leq \frac{1}{C}\overrightarrow{\mathbf{w}}\cdot\overrightarrow{\mathbf{h}} = \frac{1}{C}(C\alpha\beta - (C-1)\epsilon_3') = \alpha\beta - \frac{C-1}{C}\epsilon_3'
$$

The second-order derivative of $\log(1 + (C-1)\exp(x))$ is

$$
\frac{(C-1)\exp(x)}{(1 + (C-1)\exp(x))^2} = 1/((C-1)\exp(x) + 2 + \frac{1}{(C-1)\exp(x)}),
$$

which is $e^{-\kappa C\alpha\beta}$ for any $x \in [-\frac{C^2}{C-1}\alpha\beta, \frac{C^2}{C-1}\alpha\beta]$ for small constant $\kappa$, we denote as $O(C\alpha\beta)$ further. Therefore, the function $\log(1 + (C-1)\exp(x))$ is $\lambda$-strongly-convex for $\lambda = e^{-O(C\alpha\beta)}$ Thus, for any subset $S \subseteq [C]$, let $\delta = \frac{|S|}{C}$, by 2.1:

$$
-\frac{C}{C-1}\sum_{c\in S}\dot{\mathbf{w}}_c\tilde{\mathbf{h}}_c \leq \delta C(-\frac{1}{C-1}\overrightarrow{\mathbf{w}}\cdot\overrightarrow{\mathbf{h}}) + C\sqrt{\frac{2\epsilon_2'\delta(1-\delta)}{\lambda}}
$$

$$
\sum_{c\in S}\dot{\mathbf{w}}_c\tilde{\mathbf{h}}_c \geq \delta\overrightarrow{\mathbf{w}}\cdot\overrightarrow{\mathbf{h}} - (C-1)\sqrt{\frac{2\epsilon_2'\delta(1-\delta)}{\lambda}}
$$

$$
\begin{aligned}
\sum_{c\in S}\alpha_c\beta_c &= \sum_{c\in[C]}\alpha_c\beta_c - \sum_{c\notin S}\alpha_c\beta_c \\
&\leq \sum_{c\in[C]}\alpha_c\beta_c - \sum_{c\notin S}\dot{\mathbf{w}}_c\tilde{\mathbf{h}}_c \\
&\leq C\alpha\beta - \sum_{c\notin[C]-S}\dot{\mathbf{w}}_c\tilde{\mathbf{h}}_c \\
&\leq C\alpha\beta - (1-\delta)\overrightarrow{\mathbf{w}}\cdot\overrightarrow{\mathbf{h}} + (C-1)\sqrt{\frac{2\epsilon_2'\delta(1-\delta)}{\lambda}}
\end{aligned}
$$

Let $\alpha_c = \sqrt{\frac{1}{N}\sum_{i=1}^{N}\|\mathbf{h}_{c,i}\|^2}$ and $\beta_c = \|\dot{\mathbf{w}}_c\|$. Note that since $-\frac{1}{C-1}\overrightarrow{\mathbf{w}}\cdot\overrightarrow{\mathbf{h}} = -\frac{C}{C-1}\alpha\beta + \epsilon_3'$, there is $\overrightarrow{\mathbf{w}}\cdot\overrightarrow{\mathbf{h}} = C\alpha\beta - (C-1)\epsilon_3'$. Therefore,

$$\sum_{c\in S}\dot{\mathbf{w}}_c\tilde{\mathbf{h}}_c \geq \delta C\alpha\beta - \delta(C-1)\epsilon_3' - (C-1)\sqrt{\frac{2\epsilon_2'\delta(1-\delta)}{\lambda}}$$

$$\sum_{c\in S}\alpha_c\beta_c \leq \delta C\alpha\beta + (1-\delta)(C-1)\epsilon_3' + (C-1)\sqrt{\frac{2\epsilon_2'\delta(1-\delta)}{\lambda}}$$

Therefore, there are at most $\delta C$ classes for which

$$\dot{\mathbf{w}}_c\tilde{\mathbf{h}}_c \leq \alpha\beta - \frac{(C-1)}{C}\epsilon_3' - \frac{C-1}{C}\sqrt{\frac{2\epsilon_2'(1-\delta)}{\delta\lambda}} \tag{19}$$

and also there are at most $\delta C$ classes for which

$$\alpha_c\beta_c \geq \alpha\beta + \frac{(1-\delta)(C-1)}{\delta C}\epsilon_3' + \frac{C-1}{C}\sqrt{\frac{2\epsilon_2'(1-\delta)}{\delta\lambda}} \tag{20}$$

Thus, for at least $(1-2\delta)C$ classes, there is

$$\frac{\dot{\mathbf{w}}_c\tilde{\mathbf{h}}_c}{\alpha_c\beta_c} \geq 1 - \left(\frac{C-1}{C\alpha\beta}\right)\left(\frac{\epsilon_3'}{\delta} - 2\sqrt{\frac{2\epsilon_2'(1-\delta)}{\delta\lambda}}\right) \tag{21}$$

By setting $\epsilon_2' = \epsilon$ and $\epsilon_3' = 0$, we get the following upper bound on

$$\cos_{\angle}(\dot{\mathbf{w}}_c, \tilde{\mathbf{h}}_c) \geq 1 - 2\sqrt{\frac{2\epsilon(1-\delta)}{\delta\lambda}}$$

Using $\lambda = e^{-O(C\alpha\beta)}$ gives the NC3 bound in the theorem:

$$\cos_{\angle}(\dot{\mathbf{w}}_c, \tilde{\mathbf{h}}_c) \geq 1 - 2\sqrt{\frac{2\epsilon(1-\delta)e^{O(C\alpha\beta)}}{\delta}}$$

Therefore, applying lemma A.4 these classes, there is

$$intra_c \geq 1 - 4\left(\frac{C-1}{C\alpha\beta}\right)\left(\frac{\epsilon_3'}{\delta} - 2\sqrt{\frac{2\epsilon_2'(1-\delta)}{\delta\lambda}}\right)$$

Assuming that $\epsilon \ll 1$, then $\epsilon \ll \sqrt{\epsilon}$. Therefore, then worst case bound when $\epsilon \geq \epsilon_2' + \frac{\gamma'}{1+\gamma'}\epsilon_3'$ is achieved when $\epsilon_2' = \epsilon$:

$$intra_c \geq 1 - 8\left(\frac{C-1}{C\alpha\beta}\right)\sqrt{\frac{2\epsilon(1-\delta)}{\delta\lambda}}$$

Plug in $\lambda = \exp(-O(C\alpha\beta))$ and with simplification we get:

$$intra_c \geq 1 - \frac{(C-1)}{C\alpha\beta}\sqrt{\exp(O(C\alpha\beta))\frac{128\epsilon(1-\delta)}{\delta}} = 1 - O\left(\frac{e^{O(C\alpha\beta)}}{\alpha\beta}\sqrt{\frac{\epsilon}{\delta}}\right)$$

Now consider the inter-class cosine similarity. Let $m_c = -\frac{C}{C-1}\dot{\mathbf{w}}_c\tilde{\mathbf{h}}_c$, by Lemma A.2 we know that for any set $S$ of $\delta(C-1)$ classes in $[C]-\{c\}$, using the definition that $\dot{\mathbf{w}}_c = \mathbf{w}_c - \tilde{\mathbf{w}}$ there is

$$\sum_{c'\in S}(\dot{\mathbf{w}}_{c'} - \dot{\mathbf{w}}_c)\tilde{\mathbf{h}}_c = \sum_{c'\in S}(\mathbf{w}_{c'} - \mathbf{w}_c)\tilde{\mathbf{h}}_c \leq \delta(C-1)m_c + (C-1)\sqrt{\frac{2\delta\epsilon_{1,c}'}{\exp(m_c)}}$$

Therefore, for at least $(1-\delta)(C-1)$ classes, there is

$$(\dot{\mathbf{w}}_{c'} - \dot{\mathbf{w}}_c)\tilde{\mathbf{h}}_c \leq m_c + \sqrt{\frac{2\epsilon_{1,c}'}{\exp(m_c)\delta}} = -\frac{C}{C-1}\dot{\mathbf{w}}_c\tilde{\mathbf{h}}_c + \sqrt{\frac{2\epsilon_{1,c}'}{\exp(m_c)\delta}} \tag{22}$$

$$\dot{\mathbf{w}}_{c'}\tilde{\mathbf{h}}_c \leq -\frac{1}{C-1}\dot{\mathbf{w}}_c\tilde{\mathbf{h}}_c + \sqrt{\frac{2\epsilon_{1,c}'}{\exp(m_c)\delta}} \tag{23}$$

Combining with equation 19 equation 20, we get that there are at least $(1 - 2\delta)C \times (1 - 3\delta)C \geq (1 - 5\delta)C^2$ pairs of classes $c, c'$ that satisfies the following: for both $c$ and $c'$, equations equation 19 equation 20 are not satisfied (i.e. satisfied in reverse direction), and equation 22 is satisfied for the pair $c', c$. Note that this implies

$$m_c = -\frac{C}{C-1}\dot{\mathbf{w}}_c\tilde{\mathbf{h}}_c \leq -\frac{C}{C-1}\alpha\beta + \epsilon'_3 + \sqrt{\frac{2\epsilon'_2(1-\delta)}{\delta\lambda}}$$

and

$$\dot{\mathbf{w}}_{c'}\tilde{\mathbf{h}}_c \leq -\frac{\alpha\beta}{C-1} + \frac{1}{C}(\epsilon'_3 + \sqrt{\frac{2\epsilon'_2(1-\delta)}{\delta\lambda}}) + \sqrt{\frac{2\epsilon'_{1,c}}{\exp(m_c)\delta}}$$

We now seek to simplify the above bounds using the constraint that $\epsilon \geq \frac{1}{C}\sum_{c=1}^{C}\frac{\epsilon'_{1,c}}{C} + \epsilon'_2 + \frac{\gamma'}{1+\gamma'}\epsilon'_3$. Note that $\epsilon \ll \sqrt{\epsilon}$, and both $\lambda$ and $\exp(m_c)$ are $\exp(-O(C\alpha\beta))$, therefore, we can achieve the maximum bound by setting $\epsilon'_{1,c} = \epsilon$,

$$\dot{\mathbf{w}}_{c'}\tilde{\mathbf{h}}_c \leq -\frac{\alpha\beta}{C-1} + \exp(O(C\alpha\beta))\sqrt{\frac{2\epsilon}{\delta}}$$

Similarly, we can achieve the smallest bound on $\alpha_c\beta_c$ (the reverse of equation 20)by setting $\epsilon'_2 = \epsilon$ and using $\lambda = \exp(-O(\alpha\beta))$ we get for both $c$ and $c'$

$$\alpha_c\beta_c \leq \alpha\beta + \exp(O(\alpha\beta))\sqrt{\frac{2\epsilon}{\delta}}$$

and achieve the largest bound on $\dot{\mathbf{w}}_c\tilde{\mathbf{h}}_c$ (the reverse of equation 19) by setting $\epsilon'_2 = \epsilon$ we get for both $c$ and $c'$:

$$\dot{\mathbf{w}}_c\tilde{\mathbf{h}}_c \leq \alpha\beta - \exp(O(C\alpha\beta))\sqrt{\frac{2\epsilon}{\delta}}$$

Therefore, we can apply Lemma A.5 with $\alpha = \alpha_c$, $\beta = \beta_c$, $\epsilon' = \alpha_c\beta_c - \dot{\mathbf{w}}_c\tilde{\mathbf{h}}_c \leq 2\exp(O(C\alpha\beta))\sqrt{\frac{2\epsilon}{\delta}}$ bound to get:

$$\cos_\angle(\dot{\mathbf{w}}_{c'}, \tilde{\mathbf{h}}_c) \leq -\frac{1}{C-1} + \frac{C}{C-1}\frac{\exp(O(C\alpha\beta))}{\alpha\beta}\sqrt{\frac{2\epsilon}{\delta}} + 4(\frac{2\exp(O(C\alpha\beta))}{\alpha\beta}\sqrt{\frac{2\epsilon}{\delta}})^{1/3}$$

$$\leq -\frac{1}{C-1} + O(\frac{e^{O(C\alpha\beta)}}{\alpha\beta}(\frac{\epsilon}{\delta})^{1/6})$$

Where the last inequality is because $\frac{e^{O(C\alpha\beta)}}{\alpha\beta} > 1, \frac{\epsilon}{\delta} < 1$. Finally, we derive an upper bound on $\cos_\angle(\tilde{\mathbf{h}}_{c'}, \tilde{\mathbf{h}}_c)$ and thus intra-class cosine similarity by combining the above bounds. Note that for $\frac{\pi}{2} < a < \pi$ and $0 < b < \frac{pi}{2}$ we have:

$$\cos(a - b) = \cos(a)\cos(b) + \sin(a)\sin(b)$$
$$\leq \cos(a) + \sin(b)$$
$$\leq \cos(a) + \sqrt{1 - \cos^2(b)}$$
$$\leq cos(a) + \sqrt{2(1 - \cos(b))}$$

by equation 21 we get that

$$\cos_\angle(\dot{\mathbf{w}}_{c'}, \tilde{\mathbf{h}}_{c'}) \geq 1 - (\frac{C-1}{C\alpha\beta})(\frac{\epsilon'_3}{\delta} - 2\sqrt{\frac{2\epsilon'_2(1-\delta)}{\delta\lambda}}) \geq 1 - \frac{\exp(O(C\alpha\beta))}{\alpha\beta}\sqrt{\frac{2\epsilon}{\delta}}$$

Therefore,

$$\cos_\angle(\tilde{\mathbf{h}}_{c'}, \tilde{\mathbf{h}}_c) \leq \cos_\angle(\dot{\mathbf{w}}_{c'}, \tilde{\mathbf{h}}_c) + \sqrt{2(1 - \cos_\angle(\dot{\mathbf{w}}_{c'}, \tilde{\mathbf{h}}_{c'}))}$$

$$\leq -\frac{1}{C-1} + \frac{C}{C-1}\frac{\exp(O(C\alpha\beta))}{\alpha\beta}\sqrt{\frac{2\epsilon}{\delta}} + 4(\frac{2\exp(O(C\alpha\beta))}{\alpha\beta}\sqrt{\frac{2\epsilon}{\delta}})^{1/3} + \sqrt{\frac{\exp(O(C\alpha\beta))}{\alpha\beta}\sqrt{\frac{2\epsilon}{\delta}}}$$

$$= -\frac{1}{C-1} + O(\frac{e^{O(C\alpha\beta)}}{\alpha\beta}(\frac{\epsilon}{\delta})^{1/6})$$

Since $\|\tilde{\tilde{\mathbf{h}}}_c\| \leq 1$, there is

$$\tilde{\tilde{\mathbf{h}}}_{c'} \cdot \tilde{\tilde{\mathbf{h}}}_c = \|\tilde{\tilde{\mathbf{h}}}_{c'}\|\|\tilde{\tilde{\mathbf{h}}}_c\| \cos_\angle(\tilde{\tilde{\mathbf{h}}}_{c'}, \tilde{\tilde{\mathbf{h}}}_c) \leq -\frac{1}{C-1} + O(\frac{e^{O(C\alpha\beta)}}{\alpha\beta}(\frac{\epsilon}{\delta})^{1/6})$$

Applying A.3 shows the bound on inter-class cosine similarity. Note that although this bound holds only for $1 - 5\delta$ fraction of pairs of classes, changing the fraction to $1 - \delta$ only changes $\delta$ by a constant factor and does not affect the asymptotic bound.

## A.4 PROOF OF THEOREM 2.2

**Theorem 2.2.** *For an neural network classifier without bias terms trained on a dataset with the number of classes $C \geq 3$ and samples per class $N \geq 1$, under the following assumptions:*

1. *The network contains an batch normalization layer without bias term before the final layer with trainable weight vector $\boldsymbol{\gamma}$;*

2. *The layer-peeled regularized cross-entropy loss with weight decay $\lambda < \frac{1}{\sqrt{C}}$*

$$\mathcal{L}_{\text{reg}} = \frac{1}{CN} \sum_{c=1}^{C} \sum_{i=1}^{N} \mathcal{L}_{\text{CE}}\left(f(\boldsymbol{x}_{c,i}; \boldsymbol{\theta}), \boldsymbol{y}_c\right) + \frac{\lambda}{2}(\|\boldsymbol{\gamma}\|^2 + \|\mathbf{W}\|_F^2)$$

*satisfies $\mathcal{L}_{\text{reg}} \leq m_{reg} + \epsilon$ for small $\epsilon$; where $m_{reg}$ is the minimum achievable regularized loss*

*then for at least $1 - \delta$ fraction of all classes , with $\frac{\epsilon}{\delta} \ll 1$, $\epsilon < \lambda$ and for small constant $\kappa > 0$ there is*

$$intra_c \geq 1 - \frac{C-1}{C\alpha\beta}\left(\frac{Ce}{\lambda}\right)^{\frac{\kappa C}{2}} \sqrt{\frac{128\epsilon(1-\delta)}{\delta}} = 1 - O\left(\left(\frac{C}{\lambda}\right)^{O(C)} \sqrt{\frac{\epsilon}{\delta}}\right),$$

*and also for a cosine similarity representation of NC3 in Papyan et al. (2020):*

$$\cos_\angle(\dot{\mathbf{w}}_c, \tilde{\mathbf{h}}_c) \geq 1 - 2\left(\frac{Ce}{\lambda}\right)^{\frac{\kappa C}{2}} \sqrt{\frac{2\epsilon(1-\delta)}{\delta}} = 1 - O\left(\left(\frac{C}{\lambda}\right)^{O(C)} \sqrt{\frac{\epsilon}{\delta}}\right),$$

*and for at least $1 - \delta$ fraction of all pairs of classes $c, c'$, with $\frac{\epsilon}{\delta} \ll 1$, there is*

$$inter_{c,c'} = -\frac{1}{C-1} + O(\left(\frac{C}{\lambda}\right)^{O(C)}(\frac{\epsilon}{\delta})^{1/6})$$

*Proof.* Let $\boldsymbol{\gamma}^*$ and $\boldsymbol{W}^*$ be the weight vector and weight matrix that achieves the minimum achievable regularized loss. Let $\alpha = \|\boldsymbol{\gamma}\|$ and $\beta = \frac{\|\boldsymbol{W}\|_F}{\sqrt{C}}$, and $\alpha^*$ and $\beta^*$ represent the values at minimum loss accordingly. According to Proposition 2.1, we know that $\sqrt{\frac{1}{N}\sum_{i=1}^{N}\|\mathbf{h}_i\|_2^2} = \|\boldsymbol{\gamma}\|_2 = \alpha$. From Theorem 2.1 we know that, under fixed $\alpha\beta$, the minimum achievable unregularized loss is $\log(1 + (C-1)\exp(-\frac{C}{C-1}\alpha\beta))$. Since only the product $\gamma = \alpha\beta$ is of interest to Theorem 2.1, we make the following observation:

$$\begin{aligned}
\mathcal{L}_{\text{reg}} &= \frac{1}{CN} \sum_{c=1}^{C} \sum_{i=1}^{N} \mathcal{L}_{\text{CE}}\left(f(\boldsymbol{x}_{c,i}; \boldsymbol{\theta}), \boldsymbol{y}_c\right) + \frac{\lambda}{2}(\|\boldsymbol{\gamma}\|^2 + \|\mathbf{W}\|_F^2) \\
&\geq \log(1 + (C-1)\exp(-\frac{C}{C-1}\alpha\beta)) + \frac{\lambda}{2}(\alpha^2 + C\beta^2) \\
&\geq \log(1 + (C-1)\exp(-\frac{C}{C-1}\gamma)) + \sqrt{C}\lambda\gamma \\
&\geq \min_\gamma \log(1 + (C-1)\exp(-\frac{C}{C-1}\gamma)) + \sqrt{C}\lambda\gamma
\end{aligned}$$

Now we analyze the properties of this function. For simplicity, we combine $\sqrt{C}\lambda$ into $\lambda$ in the following proposition:

**Proposition A.1.** *The function $f_\lambda(\gamma) = \log\left(1 + (C-1)\exp(-\frac{C}{C-1}\gamma)\right) + \lambda\gamma$ have minimum value*

$$f_\lambda(\gamma^*) = \log(1 - \frac{C-1}{C}\lambda) + \frac{C-1}{C}\lambda\log\left(\frac{C-(C-1)\lambda}{\lambda}\right)$$

*achieved at $\gamma^* = O(\log(\frac{1}{\lambda}))$ for $\lambda < 1$. Furthermore, for any $\gamma$ such that $f_\lambda(\gamma) - f_\lambda(\gamma^*) \le \epsilon \ll \lambda$, there is $|\gamma - \gamma^*| \le \sqrt{O(1/\lambda)\epsilon}$*

*Proof.* Consider the optimum of the function by setting the derivative to 0:

$$g'_\lambda(\gamma^*) = -\frac{C}{C-1}\frac{(C-1)\exp(-\frac{C}{C-1}\gamma^*)}{\left(1+(C-1)\exp(-\frac{C}{C-1}\gamma^*)\right)} + \lambda = 0$$

$$\frac{C-1}{C}\lambda = 1 - \frac{1}{1+(C-1)\exp(-\frac{C}{C-1}\gamma^*)}$$

$$1 + (C-1)\exp(-\frac{C}{C-1}\gamma^*) = \frac{1}{1-\frac{C-1}{C}\lambda}$$

$$\gamma^* = \frac{C-1}{C}\log\left(\frac{C-(C-1)\lambda}{\lambda}\right) < \log(\frac{C}{\lambda})$$

Plugging in $\gamma^* = \frac{C-1}{C}\log\left(\frac{C-(C-1)\lambda}{\lambda}\right)$ to the original formula we get:

$$f_\lambda(\gamma^*) = \log(1 - \frac{C-1}{C}\lambda) + \frac{C-1}{C}\lambda\log\left(\frac{C-(C-1)\lambda}{\lambda}\right)$$

Note that since $\gamma \ge 0$, the optimum point is only positive when $\lambda \le 1$.

Now consider the case where the loss is near-optimal and $\gamma = \gamma^* + \epsilon'$ for $\epsilon' \ll 1$:

$$\log\left(1 + (C-1)\exp(-\frac{C}{C-1}(\gamma^* + \epsilon'))\right) + \lambda(\gamma^* + \epsilon')$$

$$\ge \log\left(1 + (C-1)\exp(-\frac{C}{C-1}\gamma^*)(1 - \frac{C}{C-1}\epsilon' + \frac{\epsilon'^2}{2})\right) + \lambda(\gamma^* + \epsilon')$$

$$\ge \log\left(1 + (C-1)\exp(-\frac{C}{C-1}\gamma^*)\right) + \frac{(C-1)\exp(-\frac{C}{C-1}\gamma^*)}{\left(1+(C-1)\exp(-\frac{C}{C-1}\gamma^*)\right)}(-\frac{C}{C-1}\epsilon' + \frac{\epsilon^2}{2}) + \lambda(\gamma^* + \epsilon')$$

By definition of $\gamma^*$ as the optimal $\gamma$, the first-order term w.r.t. $\epsilon'$ must cancel out. Also, by plugging in $\gamma^*$, the coefficient of $\frac{\epsilon'^2}{2}$ is $\frac{C-1}{C}\gamma$. Therefore,

$$\log\left(1 + (C-1)\exp(-\frac{C}{C-1}(\gamma^* + \epsilon'))\right) + \lambda(\gamma^* + \epsilon')$$

$$\le \log\left(1 + (C-1)\exp(-\frac{C}{C-1}\gamma^*)\right) + \lambda\gamma^* + \frac{C-1}{C}\lambda\epsilon'^2$$

Conversely, for any $\epsilon \ll 1$ for which $g(\gamma) \le g(\gamma^*) + \epsilon$, there must be $|\gamma - \gamma^*| \le \sqrt{\frac{C\epsilon}{(C-1)\lambda}\epsilon}$ $\qquad\square$

Thus, the minimum achievable value of the regularized loss is

$$m_{\text{reg}} = \log(1 - \frac{C-1}{\sqrt{C}}\lambda) + \frac{C-1}{\sqrt{C}}\lambda\log\left(\frac{\sqrt{C}}{\lambda} - (C-1)\right)$$

Now, consider any $\mathbf{W}$ and $\gamma$ that achieves near-optimal regularized loss $\mathcal{L}_{\text{reg}} = m_{\text{reg}} + \epsilon$ for very small $\epsilon$. Recall that $\alpha = \|\boldsymbol{\gamma}\|$, $\beta = \frac{\|\mathbf{W}\|_F}{\sqrt{C}}$, $\gamma = \alpha\beta$. According to Proposition A.1 we know that $|\gamma - \gamma^*| \le \sqrt{\frac{C\epsilon}{(C-1)\lambda}}$. Therefore, $\gamma \le \gamma^* + \sqrt{\frac{C\epsilon}{(C-1)\lambda}} = \log(C/\lambda) + \sqrt{\frac{C\epsilon}{(C-1)\lambda}}$. Also, note that $\mathcal{L}_{\text{reg}} - f_{\sqrt{C}\lambda}(\gamma) \le \mathcal{L}_{\text{reg}} - f_{\sqrt{C}\lambda}(\gamma^*) = \epsilon$, where $f_{\sqrt{C}\lambda}(\gamma)$ is the minimum unregularized loss according to Theorem 2.1. Therefore, we can apply Theorem 2.1 with $\alpha\beta = \gamma < \log(C/\lambda) + \sqrt{\frac{C\epsilon}{(C-1)\lambda}}$ and the same $\epsilon$ to get the results in the theorem.

$\qquad\square$

# B    ADDITIONAL EXPERIMENTS

This section presents more comprehensive experimental results that support our conclusion.

## B.1    EXPERIMENTS ON SYNTHETIC DATASETS

### B.1.1    EXPERIMENTAL SETUP

Our results in the main paper show the intra-class and inter-class cosine similarity results for 3-layer and 6-layer multi-layer perceptrons on the conic hull datasets. To further investigate the effect of Batch Normalization and Weight Decay on more complex synthetic datasets, we randomly initialize the weight of a 3-layer and 6-layer MLP network with the same architecture as the model used in training. We then sample random vectors from a standard Gaussian distribution and use the index of the maximum element of the output of the randomly initialized MLP as the label. The corresponding datasets generated using 3-layer and 6-layer randomly initialized models are called MLP3 and MLP6 datasets, respectively. Our intuition is that by generating data using a randomly initialized network, we can control the complexity of the underlying distribution, unlike vision datasets such as MNISTLeCun et al. (2010) and CIFAR10Krizhevsky (2009) where the distribution cannot be strictly defined. We run our experiments on models of 3 different depths (3, 6, 9). For each model depth, we create a version with batch normalization between each adjacent hidden layer and a version without any batch normalization. We used 8000 training samples sampled from each distribution (conic hull dataset, MLP3 dataset, and MLP6 dataset). Other hyperparameters are the same as described in the main paper. All experiments in this subsection are performed on Google Colab.

### B.1.2    EXPERIMENTAL RESULTS

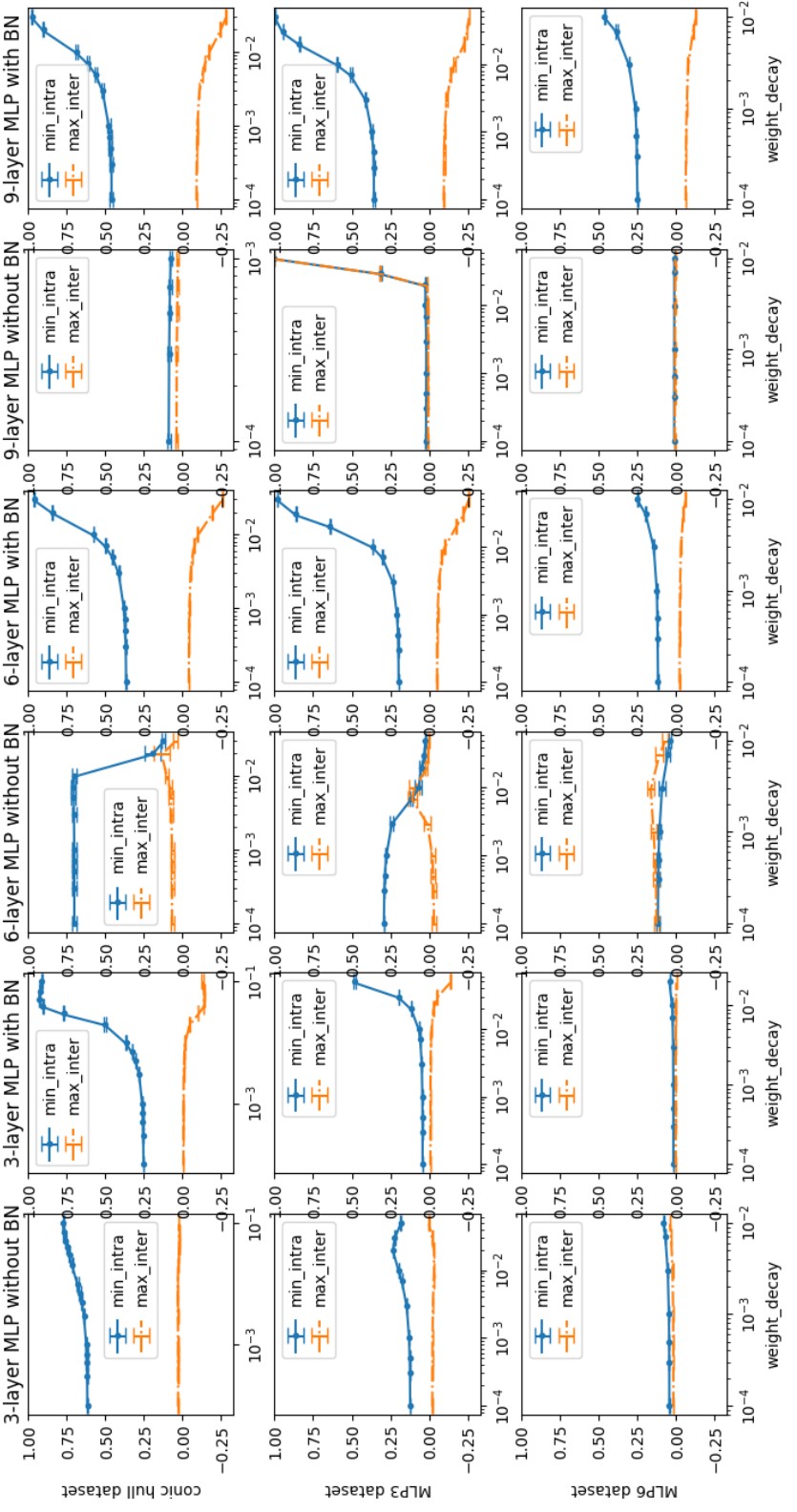

Figure 4: Experimental Results for Synthetic datasets using Multi-layer Perceptron (MLP) models of different depths with and without batch normalization. Note that the models did not converge and achieve perfect accuracy in all cases but we include them for completeness. Notably, the SGD algorithm is not able to efficiently train the 6-layer and 9-layer MLP network without batch normalization in our setting but is able to converge on the batch-normalized model. The 3-layer MLP is also unable to classify the MLP3 and MLP6 dataset perfectly, which is demonstrated by the intra-class and inter-class cosine similarity being not far apart.

## B.2 EXPERIMENTS WITH REAL-WORLD DATASETS

### B.2.1 EXPERIMENTAL SETUP

Our next set of experiments trains popular computer vision neural network models VGG-11, VGG-11+BN, VGG-19, VGG-19+BN, and ResNet-18 to investigate the effect of batch normalization and weight decay on the emergence of Neural Collapse. Note that the VGG-11+BN, VGG-19+BN, and ResNet18 models contain batch normalization layers, while VGG11 and VGG19 networks do not contain any batch normalization. We perform the experiments on the vision datasets MNIST (LeCun et al. (2010)), CIFAR10, and CIFAR100 (Krizhevsky (2009)). For all VGG models, we train for 200 epochs on each dataset, while we only train for 100 epochs for the ResNet18 model (due to resource constraints). For ResNet, we also used a subset of 8000 training samples for the CIFAR10 and MNIST datasets. All other hyperparameters are the same as in the main paper.

### B.2.2 EXPERIMENTAL RESULTS

See Figure B.2.2. We note that in certain experiments using models with Batch Normalization, the inter-class cosine similarity may begin to rise at high weight-decay levels once the intra-class cosine similarity of all classes reaches close to one. This can be explained by under-fitting at high weight decay values. More specifically, because of the regularization effect of high weight decay, the model cannot properly interpolate the training data and attain near-optimal training loss, which is a necessary condition for Neural Collapse and our theoretical analysis.

## C COMPARISON WITH OTHER THEORETICAL WORKS ON THE EMERGENCE OF $\mathcal{NC}$

| | MSE | CE | Reg. | Norm. | Opt. | Landscape | Near-Opt. |
|---|---|---|---|---|---|---|---|
| Ji et al. (2022) | | ✓ | | | ✓* | ✓* | |
| Zhu et al. (2021) | | ✓ | ✓ | | ✓ | ✓ | |
| Lu & Steinerberger (2022) | | ✓ | | ✓ | ✓ | | |
| Poggio & Liao (2020) | ✓ | | | ✓ | ✓ | ✓ | |
| Tirer & Bruna (2022) | ✓ | | ✓ | | ✓ | | |
| Súkeník et al. (2023) | ✓ | | ✓ | | ✓ | | |
| Han et al. (2022) | ✓ | | ✓ | | ✓ | ✓ | |
| Yaras et al. (2022) | | ✓ | | ✓ | ✓ | ✓ | |
| E & Wojtowytsch (2022) | | ✓ | | ✓ | ✓ | | |
| This Work | | ✓ | ✓ | ✓ | ✓ | | ✓ |

Table 1: Comparison with existing theoretical works on the emergence of $\mathcal{NC}$. "Reg." denotes weight or feature norm regularization assumption, "Norm." denotes weight or feature norm constraint/normalization, "Opt." denotes optimality conditions, and "Landscape" denotes landscape or gradient flow analysis. * Shows the direction of gradient flow as it tends towards infinity without normalization/regularization.

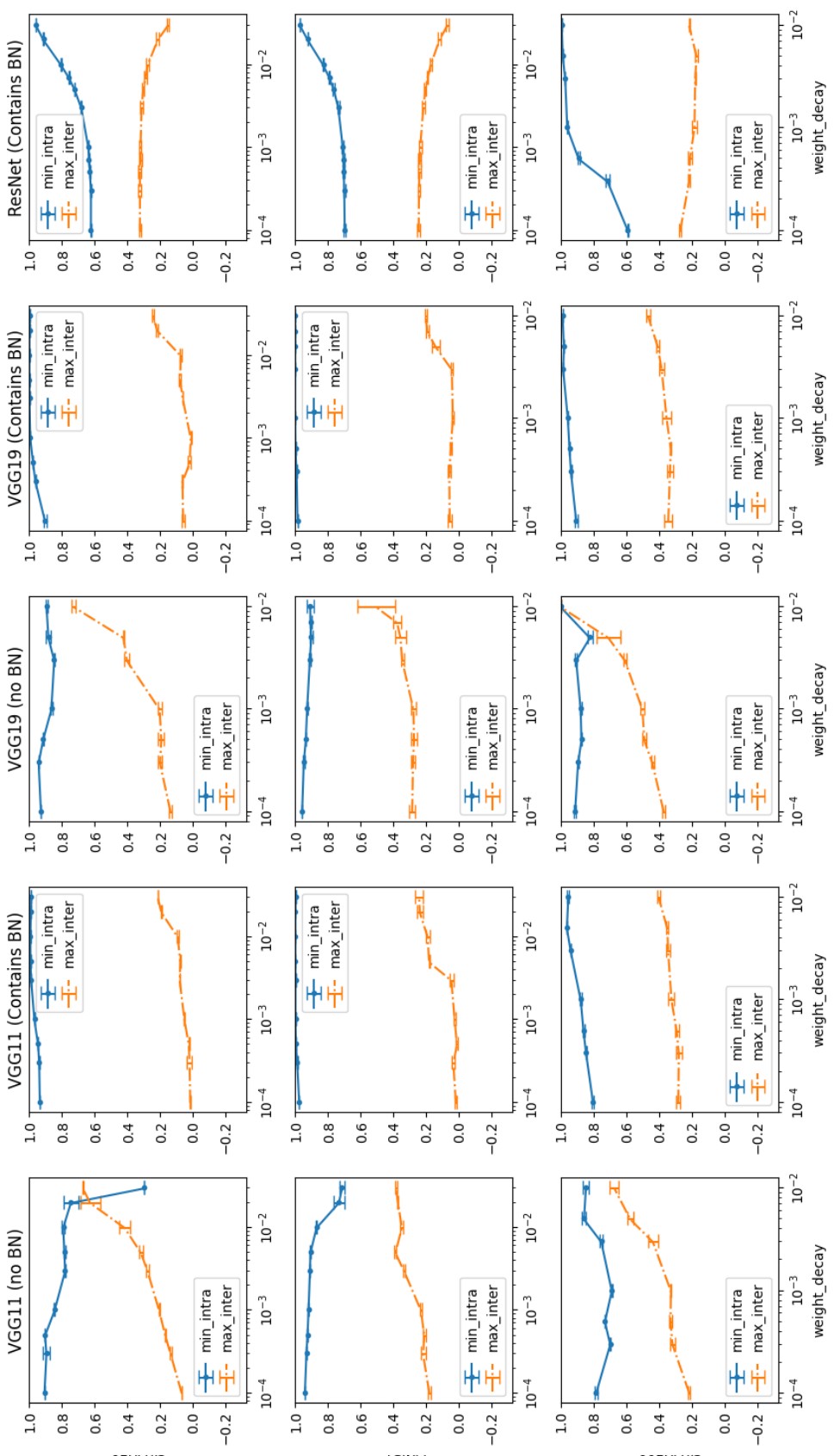

Figure 5: Experimental Results for Real-World Computer Vision Datasets and standard Computer Vision Models with and without batch normalization.

