# OpenReview forum: "Towards Understanding Neural Collapse: The Effects of Batch Normalization and Weight Decay"
_ICLR.cc/2024/Conference — Submitted to ICLR 2024_

### Official Review · Reviewer_PiTj · 2023-10-17

**Soundness:** 3 good
**Presentation:** 3 good
**Contribution:** 3 good
**Rating:** 5
**Confidence:** 4

**Summary:**

The paper discusses neural collapse and links it to batch normalization and weight decay regularization, in settings where the empirical cross entropy is close to its optimal value. Both theoretical and empirical evidence is presented for this connection.

**Strengths:**

The paper deals with an interesting and timely topic and contributes to a better understanding of the phenomenon of neural collapse. The logical flow of arguments is clear, and the paper is quite accessible (thanks, e.g., to the sketch of the proof). The empirical evidence, at least for smaller architectures and synthetic datasets, seems to support the claims made based on theoretical considerations. This is a major strength of the paper, together with the theoretical analysis.

**Weaknesses:**

Some weaknesses currently prevent me from giving a better score, and I would appreciate the authors opinion on these:
* For the theoretical analysis, the layer-peeled model is a very specific setting. I understand that narrow settings may be required to obtain rigorous results. I would like to understand, however, the rationale behind this choice and how the authors believe that the results can generalize (also theoretically) to more general settings.
* At the beginning of Section 2.2, the authors claim that quantities proposed in the past to quantify NC fail to deliver insights whenever the "values are non-zero". Why is that the case? And how does cosine similarity excel in this regard? To my basic understanding, a measure such as SNR (which I take to be the ratio between inter-class distances over intra-class distances) seems quite intuitive (and also accounts for vector lengths, which the proposed cosine similarity does not). To some extent, I would even believe that SNR allows a similar theoretical analysis as cosine similarity (assuming that SNR also depends on Euclidean distances).
* Some experimental details are not clear (see questions below), I would appreciate clarifications. Most importantly, what are the training and test errors for the settings in Fig. 1 and 2. I would appreciate if these curves could be plotted on top of the existing ones, to get a better understanding of the training success.
* Another shortcoming of the paper is that the results for the larger architectures in Fig. 2 are not as clear; VGG11, for example, seems not to exhibit NC, at least not using the measures that are proposed. This limits the contribution of the paper slightly, as the phenomenon of NC is either not as prevalent as claimed, or the proposed measures are inadequate of capturing this phenomenon in interesting settings (i.e. for large architectures).
* Connected to the previous point, the authors state that cosine similarity describes "necessary conditions for the core observations of NC". Hence, if NC occurs, this should be reflected in the measures (which suggests that the results in Fig. 2 do not exhibit NC). In contrast, even if the proposed measures behave as expected under NC (as in Fig. 1), there is no guarantee that NC actually occurred. This is some, if only a minor, shortcoming of the proposed measure. I acknowledge, however, that the authors are aware of this limitation and I agree that this does not affect the validity and usefulness of the theoretical results.

**Questions:**

* Why does the inter-class cosine similarity under the NC setting (NC2 on top of page 5) not depend on $d$? Is it implicitly assumed that $d>C$?
* Is the $\beta$ in Th. 2.1 the same as the $\beta$ in the definition of BN? Is the $\lambda$ in Lemma 2.1 the same as the regularization parameter?
* Does "unbiased BN" refer to $\beta=0$?
* In Section 3.1, what is the number of classes $C$? What is the dimension of the penultimate layer $d$? In Section 3.2, what is the dimension of the penultimate layer $d$?

---

> ### Author Response · Authors · 2023-11-20
> **Reply to reviewer PiTj**
>
> We sincerely appreciate your recognition of the strengths of our paper, especially in its contribution to understanding neural collapse and the accessibility of our logical argumentation.
>
> ## Response to Weaknesses
>
> 1. As with prior works, we adapt the layer-peeled model due to the difficulty of analyzing the effect of multiple layers of non-linear activation functions and the variable neural network architecture possibilities of the lower layers. Suppose the activations functions of the Neural Network are linear (e.g. the identity function). In that case, our results can be relatively easily generalized to features of lower layers since the layers can be combined into a single linear transformation. This generalization to lower layers is much more difficult for non-linear activation functions and would likely require more intricate mathematical tools. This direction has recently been pursued by [1][2] for the MSE loss in the optimal case. However, the corresponding generalization for CE loss in the near-optimal case is more involved and likely to require significant future work.
> 2. We thank the reviewer for pointing out this point of potential misunderstanding. We would like to clarify our choice of using cosine similarity as a measure:
>     1. The cosine similarity measure is a widely used adopted similarity measure to compare between features of different samples in both practical feature learning and machine learning theory for its geometric interpretability. Quantitatively analyzing this measure allows our theorems to have more direct implications for researchers in such relevant fields
>     2. The cosine similarity measure allows direct value comparison between NC1 and NC2, corresponding to inter-class and intra-class cosine similarity respectively, which is a benefit over prior works. For example, in the original NC paper [1], the authors adopted 2 different measures, SNR ratio and feature mean cosine similarity, for NC1 and NC2, which cannot be directly compared since they are 2 different measures.
>     3. Connected to the previous point, while the SNR ratio compares intra-class variability and inter-class distance, a small SNR ratio does not distinguish whether both are small or both are large. In particular, the SNR ratio alone does not ensure proximity to the ETF structure. Explicitly measuring the cosine similarity for both intra-class and inter-class allows separate guarantees using the same measure that implies NC1 and NC2 (without norm equality) when combined.
>     4. Cosine similarity also allows relatively simple quantitative analysis.
>
>     We would like to note that we do not intend to decrease the value of the SNR ratio as a measure, which we’re convinced to have significant intuition and insights in other fields that are not captured by cosine similarity. However, the above reasons make us believe that cosine similarity is an appropriate measure for our purpose of quantitative analysis of Neural Collapse.
>
> 3. We thank multiple reviewers for making this suggestion and will adjust our plots accordingly. The questions are answered in the Questions section
> 4. Due to the intricacy of deep neural network training, it is hard to make certain assertions on the result of training complex networks. However, we think that the following reasons may be influential in the current results for the real-world networks and datasets:
>     1. Achieving near-1 intra-class cosine similarity is a strong requirement on the features within the same class when the vector dimension is large.
>     2. We used the *minimum* intra-class cosine similarity and *maximum* inter-class cosine similarity, which again is a strong requirement for the structure of feature vectors. This measure can be affected by outliers since there are $C(C-1)$ pairs of classes for inter-class cosine similarity
>
>     To address the second point, we’re currently running experiments to also record the average intra-class and inter-class cosine similarity for the real-world models and dataset combinations to resolve the sensitivity to outliers. We will update the paper when they’re completed.
>
>
> ## Questions
>
> 1. Yes, our theorems does assume that $d\geq C$, which is required for forming a equiangular tight frame. We thank the reviewer for pointing this out and will make this assumption explicit to avoid confusion.
> 2. The answer to both questions is No. We thank the reviewers for pointing out our reuse of variables that can cause confusion and will use different notations for different parameters in our revised version.
> 3. Yes, we would adjust the term to “BN with zero bias” to better clarify our model.
> 4. In section 3.1, the number of classes for the synthetic dataset is 4 and the dimension of the penultimate layer is 200. In section 3.2, the dimension is dependent on the specific model architecture used. For VGG models, the dimension is 4096 and for ResNet, the dimension is 512.

---

> ### Comment · Reviewer_PiTj · 2023-11-22
> **Thanks**
>
> Thanks for the clarifications, I appreciate your answers and I think that revising the paper according to them will improve the work. The main concerns, however, -- unclear picture for large architectures, unclear how the results generalize to other models than the layer-peeled model -- remain. After reading the other reviewers' comments, I have thus decided to not change my score. Although the situation is more clear to me know, given that my knowledge in the field of NC is limited, I cannot reasonably champion the paper.

---

### Official Review · Reviewer_QjJX · 2023-10-23

**Soundness:** 3 good
**Presentation:** 2 fair
**Contribution:** 2 fair
**Rating:** 5
**Confidence:** 3

**Summary:**

This paper explores the phenomenon of "Neural Collapse (NC)" in the final layer of neural network classifiers. The study investigates the relationships between batch normalization (BN), weight decay, and their proximity to NC. It introduces an NC measure using intra-class and inter-class cosine similarity and provides theoretical guarantees for the emergence of NC in scenarios with near-optimal regularized cross-entropy loss. The theoretical results highlights the importance of BN and weight decay for the emergence of NC. The empirical evidence supports these findings, suggesting that BN and appropriate weight-decay values play a significant role in the occurrence of NC in neural network models.

**Strengths:**

1. The effect of batch normalization (BN) for neural collapse (NC) makes sense to me. Previous works show that feature normalization plays an important role in NC, and BN together with $\ell_2$ weight normalization encompasses feature normalization.
2. The presentation of this paper is clear and easy to understand. The authors make a thorough comparison with previous works, highlight the contribution of this paper, and discuss the limitations and future directions.The author provides both theoretical and empirical evidence to support their claims.

**Weaknesses:**

1. The use of cosine similarity is not novel. Cosine similarity has long been used in previous works (e.g. [1]) to characterize the inter-class and intra-class feature variations. The authors should avoid saying that they "propose the cosine similarity measure", and should provide a more thorough discussion of related literatures.
2. The use of cosine similarity is not well justified. Cosine similarity can only guarantee the directions of the feature vectors are similar, but has nothing to do with the norm of the feature vector. Using it as a metric for NC is too weak.
3. The theoretical contributions are limited. The major theoretical contribution of this paper, is to establish a *near-optimal* NC configuration for a *near-optimal* CE loss. There are already various previous works proving this relationship for the exact setting (e.g. [2] ), and the paper mainly follow their proof outline. The novel component is Lemma 2.1, where the authors prove an approximate version of Jenson's inequality for strongly convex objectives. Therefore, I believe the major theoretical contribution is to directly extend previous guarantees for NC to the non-asymptotic cases. Furthermore, the exponential terms in the non-asymptotic bound implies that the result has a very high dependency on the parameter norm, which again questions whether such an extension is necessary or meaningful.

Minor issues:
1. Some terminologies are not accurate. For example, the authors repeatedly use "unbiased neural network" to refer to network architectures without bias, which causes ambiguity due to its statistical meaning.
2. The figures are blurred. The authors should use a higher resolution for figures.


[1] Kornblith S, Chen T, Lee H, et al. Why do better loss functions lead to less transferable features?

[2] Jianfeng Lu and Stefan Steinerberger. Neural collapse under cross-entropy loss.

**Questions:**

Please answer the questions in the weakness section, including the use the cosine similarity and the significance of the theories.

---

> ### Author Response · Authors · 2023-11-20
> **Response to reviewer QjJX**
>
> We thank the reviewer for acknowledging the clear presentation of our work and pointing out the effect and batch normalization and weight decay in NC. Below are our responses to the points in the weakness section.
>
> 1. We thank multiple reviewers for pointing out the potentially misleading expression of our contributions on the cosine similarity. We are aware that cosine similarity has been widely used in previous literature. Our main intent is to claim that we are novel in applying cosine similarity to the specific phenomenon of analyzing Neural Collapse, which has not been done before to the best of our knowledge at the time of writing. We will clarify our contributions with regard to this statement and discuss the related use of the cosine similarity measure for similar purposes in previous literature.
> 2. As mentioned in section 2.2, we adapt cosine similarity as a quantitative measure for NC due to the straightforward geometric meaning it provides compared to previous measures. In particular, cosine similarity is a widely adapted measure for similarity between features of different inputs, both in practical feature learning and in machine learning theory. Therefore, we believe that our measures already capture the majority of the implications of NC. The simplistic mathematical definition of the cosine similarity measure also allows for relatively straightforward quantitative analysis in the near-optimal regime that can be adopted by future works. Our main novelty lies in the quantitative analysis of the influence of various factors on the guaranteed proximity to the NC structure in the near-optimal regime. The extension of this analysis to the full predictions of NC, including norm equality, requires future work.
> 3. We think that the main doubts of the reviewer on the contribution of our theorem can be summarized as a) the asymptotic nature of our theorems, b) the exponential dependency of the theorem, and c) limited theoretical contributions. We will respond to these points separately:
>     1. We would like to note that our big-O notation is not intended to indicate that our bound only works when $\epsilon\rightarrow 0$. We addressed this in detail in the global response.
>     2. For theorem 3.1, we believe that the exponential dependence on $\alpha\beta$ does not make the theorem vacuous or meaningless. Under the presence of batch normalization and high weight decay, the value of $\alpha\beta$ are not typically large. In fact, the dependence of the bound on the weight and feature norms is the exact point made by the theorem.
>
>         For theorem 3.2, the exponential dependence on $\frac{1}{\lambda}$ may indeed be vacuous in a large number of settings. We have carefully examined our proof and found that our results can be improved to remove this exponential dependence. The newly derived bound has been noted in the global response.
>
>     3. In analyzing the near-optimal scenario, we quantitatively investigate the effects of weight decay parameters and the dependence on the loss value, offering a novel perspective on the understanding of weight decay and training loss value. Moreover, our result—focusing on the near-optimal regime with batch normalization and weight decay—more closely mirrors realistic settings, in contrast to the less realistic assumptions made in other papers analyzing NC under CE loss (e.g. [1][2]) such as exact optimality, strictly normalized feature vectors, or direct optimization of last-layer features.
>
> [1] Jinxin Zhou, Xiao Li, Tianyu Ding, Chong You, Qing Qu, and Zhihui Zhu. On the optimization landscape of neural collapse under mse loss: Global optimality with unconstrained features. arXiv preprint arXiv:2203.01238, 2022.
>
> [2] Dustin G. Mixon, Hans Parshall, and Jianzong Pi. Neural collapse with unconstrained features, 2020.
>
> ## Minor
>
> 1. We thank the reviewer for pointing out this potentially misleading expression. To avoid confusion, we will change the expression to “neural networks without the bias term”.
> 2. We will fix this in our updated manuscript.

---

### Official Review · Reviewer_TmdP · 2023-10-30

**Soundness:** 2 fair
**Presentation:** 3 good
**Contribution:** 2 fair
**Rating:** 5
**Confidence:** 3

**Summary:**

This paper theoretically investigates the relationship between BN and weight decay, and a near-optimal NC structure. As similar to most previous studies (e.g., unconstrained models Zhu et al. (2021)), this paper's theoretical analysis is limited to last-layer features but from a near-optimal perspective. The theoretical analysis indicates that BN and weight decay helps the achievement of NC in the near-optimal regime.

-----------------------
I have revised my score which more accurately reflect my current standing.

**Strengths:**

1. The writing is good, and the idea is easy to follow.
2. The perspective from Batch normalization and weight decay is novel and insightful. Theorem 2.2 demonstrates a larger weight decay coefficient $\lambda$ would result in a tighter bound over both inter- and intra-class cosine similarity. Also, the experiments partially verified their theoretical results.

**Weaknesses:**

Theoretical side:
1. The original NC phenomenon contains four aspects (conditions) but this paper only analyzes the NC1 and NC2.
2. As previous works on NC, this paper is also limited to the last-layer feature vectors.
3. The affine parameter $\gamma$ for BN is not included in theorem 3.2, it is not clear that how BN influences the inter- and intra-class cosine similarities.
4. The bias term of BN is ignored in this paper, which could be important as it would greatly influence the feature vectors' norms and further influence the bound over the cosine similarity.
Overall, though the theoretical results on weight decay is interesting, the whole contribution of this paper is marginal.

Experimental side:
1. For Sec 3., only minimum intra-class and maximum inter-class cosine similarities are analyzed. I thought the authors might want to show the hardest case for both intra-class and maximum inter-class cosine similarities. But, to be pointed, the average value should be also measured as it could capture some general information of the cosine similarities across each pair of classes.
2. In Fig 2., some experimental results cannot be predicted by their theorems. For example, for conic hull dataset and 3-layer MLP, at the same level of weight decay (e.g., $10^{-4}$), the model without BN layers holds better intra-class cosine similarities.
3. In Fig 3., I don't see the value of including the results of ResNet. There is no clear baseline for ResNet (with BN) for comparison.

Suggestions on Fig 2. and 3.
1. Figures are of low resolution.
2. The curves of Model (w/o BN) should be put into one single figure for better comparison.

**Questions:**

Is there any benefit to analyze the near-optimal regime over the optimal one?

---

> ### Author Response · Authors · 2023-11-19
> **Response to reviewer comments**
>
> We thank the reviewers for their insightful reviews!
>
> **Theoretical Side:**
>
> 1. **Response to Comment  only NC1 and NC2:**
>     - It is true that our current theorems only provide guarantees on NC1 and NC2. Compared with the full NC phenomenon, the original NC paper showed that NC4 is implied by NC1, NC2 and NC3, therefore the main missing part in our work is NC3. We did not officially include the bounds for NC3 in our main theorem since properties of class-feature vectors are typically of more practical significance than last-layer weight vectors. However, the corresponding bound for NC3 is available as an intermediate result of our proof (page 23, bottom) with the same asymptotic bound as inter-class cosine similarity. We will gladly include this as a separate result if it strengthens the contribution of the paper.
> 2. **Response to Limitation to Last-Layer Feature Vectors:**
>     - Indeed, the scope of our work, like previous studies on NC, is confined to the last-layer feature vectors. This approach aligns with the original NC claim in [1].
> 3. **Response to Affine Parameter in Theorem 3.2:**
>     - We believe the reference is to theorem 2.2. Here, the the affine parameter $\gamma$ is regularized by weight decay, as shown in $L_{\text{reg}} = L + \frac{\lambda}{2}(\|\gamma\|^2 + \|\boldsymbol{W}\|^2_F)$ in section 2.1. The theorem's conclusions are a result of the combined effect of batch normalization and weight decay, with the influence of batch normalization being implicit in the $\gamma$ parameter's effect. We will clarify this in the revised manuscript.
> 4. **Response to Ignoring the Bias Term of BN:**
>     - We appreciate the reviewer highlighting the significance of the bias term in BN, which can indeed impact vector norms and consequently the cosine similarity measures. Our current definitions of intra-class and inter-class cosine similarity assume globally centered feature vectors. To address networks with biased batch normalization, a more general cosine similarity measure for these similarities where the feature vectors are globally centered should be used which centers the feature vectors by subtracting the mean. The globally centered feature vectors is also used in the original proposal of NC in [1]. This revised definition will be included in the updated version of our paper.
>
> **Experimental Side:**
>
> 1. **Response to Analysis in Sec 3.:**
>     - We thank the reviewer for suggesting the inclusion of average values for intra-class and inter-class cosine similarities. This will be incorporated in the revised manuscript
> 2. **Response to Experimental Results in Fig 2.:**
>     - The phenomenon observed by the authors with the conic hull dataset and 3-layer MLP is indeed interesting. However, our theoretical results suggest that higher weight decay, coupled with batch normalization, leads to a greater degree of Neural Collapse at near-optimal loss. The experiments aim to demonstrate this relationship, particularly under the influence of BN and show that BN models have higher neural collapse than non-BN models at high weight decay. The comparison between models with and without BN across smaller weight decay values is beyond the scope of our current theorems since the theoretical guarantees provided by BN under very small weight decay are mostly vacuous as suggested by our theorems.
> 3. **Response to Inclusion of ResNet Results in Fig 3.:**
>     - Our objective was to illustrate that Neural Collapse levels increase with weight decay in the presence of BN, as is shown and justified in the ResNet figures even without a non-BN baseline. Training a ResNet model without BN layers to achieve near-optimal loss is challenging, which is why a direct baseline comparison was not included.
>
> **Suggestions on Fig 2. and 3.:**
>
> - We acknowledge the feedback regarding the figures. We will address these points by generating higher-resolution images and reorganizing the figures for clearer comparison in the revised manuscript.
>
> **Response to the Question:**
>
> There are 2 main benefits of analyzing the near-optimal regime
>
> 1) The exact optimal loss is rarely achieved in practical training. Therefore, while prior works provide insights into the ideal case, our theorems extend their results into more practical final losses.
>
> 2) Another more important reason is that analyzing the near-optimal regime allows quantitative analysis of the effect of different parameters on the proximity to NC. For example, while prior works have shown that feature normalization/regularization and optimal loss guarantee the exact NC structure, our theoretical results show the quantitative relationship between the guaranteed proximity to the NC structure and weight decay, which is not possible by only analyzing the optimal case.
>
> [1] Papyan, V., Han, X. Y., & Donoho, D. L. (2020). Prevalence of neural collapse during the terminal phase of deep learning training. *Proceedings of the National Academy of Sciences, 117*(40), 24652-24663.

---

> ### Comment · Reviewer_TmdP · 2023-11-22
>
> Thank you for your response, but I will keep my score.
> 1. For the experiments, in Fig. 3, there seems to be no big gap between model with and without BN.
> 2. In Fig. 2, the experiments cannot be fully predicted by the theorems proposed. The application of the theorems is limited.
> 3. Still, for theorem 2.2, the effect of $\gamma$ is implicit while a theorem involving $gamma$ explicitly is encouraged.
>
> Also, after proofreading the other reviewers' comment, I decide to retain my score.

---

> > ### Author Response · Authors · 2023-11-23
> > **Reply to reviewer Tmdp**
> >
> > We thank the reviewer for the additional comment on our reply. Below are a few comments regarding the new comments:
> >
> > 1. There is a very significant gap between models with and without BN for the worst-class intra-class and inter-class cosine similarity (i.e. the second and fourth column in Fig 3). The gap in the average case is less pronounced but is still a considerable gap, especially at high weight decay values. We believe that these are sufficient justification for our empirical conclusions.
> >
> >  This is because achieving high average intra-class cosine similarity and low average inter-class cosine similarity is a much more lenient requirement. Achieving a high value for the average measure still allows individual classes/class pairs to significantly diverge from the Neural Collapse requirement.
> >
> > 2. We do not claim that the experimental results are strictly predicted by our theorems, which provide a worst-case bound on the inter-class and intra-class cosine similarity. The exact values in the experiments are highly dependent on the training dynamics and the neural network architecture.
> > The main purpose of the experiments is to establish that BN and high weight decay achieve high levels of NC, which is verified by our experiments.
> >
> > 3. The purpose of theorem 2.2 is to establish a bound that is not dependent on the feature and weight norms and only on the weight decay parameter and the loss gap. A theorem directly incorporating the $\gamma$ parameter can be obtained by directly substituting the $\beta$ parameter with $\|\gamma\|$ (The correctness of this substitution is established by proposition 2.1). We will clarify this in future versions of the paper.

---

### Official Review · Reviewer_Svsj · 2023-11-02

**Soundness:** 2 fair
**Presentation:** 3 good
**Contribution:** 2 fair
**Rating:** 3
**Confidence:** 4

**Summary:**

This work studies the geometric structure of representations before the last layer of a deep neural network trained with cross entropy, batch normalization and weight decay. Specifically, an asymptotic bound on the average intra-class and inter-class cosine similarity in dependence of the regularization strength and the loss is proven. The theoretical results are supported by experiments on synthetic data and on MNIST / CIFAR10.

**Strengths:**

This paper considers the near-optimal regime and bounds the average intra-class and inter-class cosine similarity in dependence of the value of the loss function. It improves upon prior work which only derived the minimizers.

**Weaknesses:**

### Theory

The main theoretical results are Theorem 2.1 and 2.2. They state that if the "average last-layer feature norm and the last-layer weight matrix norm are both bounded, then achieving near-optimal loss implies that most classes have intra-class cosine similarity near one and most pairs of classes have inter-class cosine similarity near -1/(C-1)".

Qualitatively, this result is an immediate consequence of continuity of the loss function together with the fact that bounded average last-layer feature norm and bounded last-layer weight matrices implies NC.
Quantitatively, this work proves asymptotic bounds on the proximity to NC as a function of the loss. This quantitative aspect is novel. I am not convinced of its significance however, as I will outline below.
  1. The result is only asymptotic, and thus it cannot be used to estimate proximity to NC from a given loss value.
  2. The bound is used as basis to argue that *"under the presence of batch normalization
  and weight decay of the final layer, larger values of weight decay provide stronger NC guarantees in the sense that the intra-class cosine similarity of most classes is nearer to 1 and the inter-class cosine similarity of most pairs of classes is nearer to -1/(C-1)."*
This is backed up by the observation, that the bounds get more tight if the weight decay parameter $\lambda$ increases. To be more specific, Theorem 2.2 shows that if $L< L{min}+\epsilon$, then the average intra class cosine similarity is smaller than $-1/(C-1) + O(f(C,\lambda,\epsilon,\delta))$ and $f$ decreases with $\lambda$.
The problem with this argument is that the loss function itself depends on the regularization parameter $\lambda$ and so it is a-priori not clear whether values of $\epsilon$ are comparable for different $\lambda$. For example, apply this argument to the more simple loss function $L(x,\lambda)=\lambda x^2$. As $L$ is convex, it is clear that the value of $\lambda>0$ is irrelevant for the minimum and the near optimal solutions. Yet, $L(x,\lambda)<\epsilon$ implies $x^2<\epsilon/\lambda$ which decreases with $\lambda$. By the logic given in this work, the latter inequality suggests that minimizing a loss function with a larger value of $\lambda$ provides stronger guarantees for arriving close to the minimum at $0$. Clearly, this is not the case and an artifact of quantifying closeness to the loss minimum by $\epsilon$, when it should have been adjusted to $\lambda \epsilon$ instead.

I have doubts on how batch normalization is handled. As far as I see, batch normalization enters the proofs only through the condition $\sum_i \| h_i \|^2 =\| h_i \|^2$ (see Prop 2.1). However, this is only an implication and batch normalization induces stronger constraints. The theorems assume that the loss minimizer is a simplex ETF in the presence of batch normalization. This is not obvious, and neither proven nor discussed. It is also not accounted for in the part of the proof of Theorem 2.2, where the loss minimum $m_{reg}$ is derived.

### Experiments

- Theorems 2.1 and 2.2 are not evaluated empirically. It is not tested, whether the average intra / inter class cosine similarities of near optimal solutions follow the exponential dependency in $\lambda$ and the square (or sixth) root dependency on $\epsilon$ as suggested by the theorems.
- Instead, the dependency of cosine similarities at the end of training (200 epochs) on weight decay strength is evaluated. As presumed by the authors, the intra class cosine similarities get closer to the optimum, if the weight decay strength increases. Yet, there are problems with this experiment. It is inconsistent with the setting of the theory part and thus only provides limited insight on if the idealized theoretical results transfer to practice.
  1. The theory part depends only on the weight decay strength on the last layer parameters. Yet, in the experiments, weight decay is applied to all layers and its strength varies between experiments (when instead only the strength of the last layer should change).
  2. The theorems assume near optimal training loss, but training losses are not reported. Moreover, the reported cosine similarities are far from optimal (e.g. intra class is around 0.2 instead of 1) which suggests that the training loss is also far from optimal. It also suggests that the models are of too small capacity to justify the 'unconstrained-features' assumption.
  3. As (suboptimally) weight decay is applied to all layers, we would expect a large training loss and thus suboptimal cosine similarities for large weight decay parameters. Conveniently, cosine similarities for such large weight decay strengths are not reported and the plots end at a weight decay strength where cosine similarities are still close to optimal.
  4. On real-world data sets, the inter class cosine similarity increases with weight decay (even for batch norm models VGG11), disagreeing with the theoretical prediction. This observation is insufficiently acknowledged.

### General

The central question that this work wants to answer **What is a minimal set of conditions that would guarantee the emergence of NC?"** is already solved in the sense that it is known that minimal loss plus a norm constraint on the features (explicit via feature normalization or implicit via weight decay) implies neural collapse. The authors argue to add batch normalization to this list but that contradicts minimality.

The first contribution listed by the authors is not a contribution.
1. *"We propose the intra-class and inter-class cosine similarity measure, a simple and geometrically intuitive quantity that measures the proximity of a set of feature vectors to several core
structural properties of NC. (Section 2.2)"*

Cosine similarity (i.e. the normalized inner product) is a well known and an extensively used distance measure on the sphere. In the context of neural collapse, cosine similarities were already used in the foundational paper by Papyan et al. (2020) to empirically quantify closeness to NC (cf. Figure 3 in this reference) and many others.

Minor:
- There is a grammatical error in the second sentence of the second paragraph
- There is no punctuation after formulas; In the appendix, multiple rows start with a punctuation
- intra / inter is sometimes written in italics, sometimes upright
- $\beta$ is used multiply with a different meaning
- Proposition 2.1 $N$ = batch site, Theorem 2.2 $N$ = number of samples per class.
- As a consequence, it seems that $\gamma$ needs to be rescaled to account for the number of batches

**Questions:**

Is it possible to arrange features in a simplex ETF such that batch normalization is satisfied? Is it possible, if $\gamma=1$, i.e., batch normalization without affine parameters? If not, does this affect the validity of Theorem 2.2?

Minor: Why is $\epsilon/\delta \ll 1$ listed as an assumption for Theorems 2.1 and 2.2? From my understanding, the big $O$ notation already states, that the result holds for $\epsilon$ small enough?

---

> ### Author Response · Authors · 2023-11-22
> **Reply to reviewer Svsj [1/3]**
>
> We thank the reviewer for the meticulous review of the robustness of our theorems and experiments and for providing valuable and constructive advice on our experimental setup. Below are some of our responses to a few points that the reviewer has made in the review.
>
> ## Theoretical Side
>
> 1. While the bound in theorems is formulated using asymptotic notation, the use of this notation is mainly intended to remove the constants and higher-order terms to simplify the formula in the theorem. It is possible to use our result to generate predictions for given values of $\epsilon$ as long as they are small enough for higher order terms of $\epsilon$ (e.g. $\epsilon^2$) to be safely ignored, which is the case for most successful training of over-parameterized neural networks. We have placed the formula for the asymptotic bound in the global response
>
>     We would like to further note that, the main insight gained from the bounds in the theoretical result is its dependency relationship on the parameters rather than the exact value. Therefore, we believe that using the asymptotic notation is sufficient in the main paper as all the dependency relationships are captured with such notation.
>
> 2. We thank the reviewer for the meticulous review of the robustness of our second theorem. We acknowledge that the loss function of our second theorem is dependent on $\lambda$. However, we’re currently not convinced that the reviewer’s argument challenges the robustness of our theorem and would appreciate further elaborations on the argument, as outlined below:
>     1. In the example given by the authors, the objective function is strongly convex with respect to the bounded variable (i.e. x), and the convexity of the function with respect to x is directly dependent on $\lambda$. This resulted in the trivial conclusion that a larger $\lambda$ value provides stronger guarantees on the value of $x$. However, in our theorems, $\lambda$ is the coefficient of the norm of the weights and the feature vectors, and thus the loss function is strongly convex with respect to the weight and feature norms. However, the intra-class and inter-class cosine similarity that our theorem provides guarantees on is not directly affected by the norm of the feature vectors since proportionally scaling all features does not affect the cosine similarity values. Therefore, the claim that “larger weight decay results in stronger guarantees under the same $\epsilon$ value from the optimal loss” is non-trivial in our theorems, unlike the example provided by the authors to illustrate their point.
>     2. If $\epsilon$ is replaced with $\epsilon\lambda$ as the reviewers suggested, the main conclusions of our work are still upheld. Notably, the bounds in theorem 3.2 are still inversely correlated to $\lambda$ if $\epsilon$ is replaced with $\epsilon\lambda$
>
> **Reply to Concern on Batch Norm:** It is true that Batch Normalization provides stronger guarantees than what is used in theorems and may lead to tighter guarantees. However, the constraint we used is a necessary implication of Batch Normalization and additional constraints of BN thus do not influence the validity of our theorem. It may be possible that additional constraints by BN may lead to better results, but this is not within the scope of this paper. On your second point where “the theorems assume that the loss minimizer is a simplex ETF in the presence of batch normalization”, our theorems do not rely on this particular assumption. Instead, this statement can be derived from our theorem by setting $\epsilon=0$. Could you explain more on where in our proof is this assumption necessary so that we can either provide more explanation or clarification in our proof?

---

> > ### Comment · Reviewer_Svsj · 2023-11-22
> >
> > Thank you for your response. I am confident that a thorough revision of the manuscript that incorporates your responses to me and the other reviewers will lead to major improvements. However, as the necessary modifications are too extensive, and as I am still not convinced about the conclusions from the theory, I will not raise my score and recommend a resubmission instead.
> >
> > Some comments:
> >
> > ### Theoretical side:
> >
> > 1. Thank you adding the 'non-asymptotic bound'. Yet, it appears to be still asymptotic as $\epsilon^2$ terms are ignored. It would be interesting to see, which values of  $\epsilon$ do are small enough to ignore the $\epsilon^2$ terms and whether such values are achieved in experiments.
> >
> > 2. I am still not convinced about the conclusions from Theorem 2.2 and that $\epsilon$ needs to be adjusted to the loss function. Obviously, the suggestion of replacing $\epsilon$ with $\lambda \epsilon$ was specific to $L = \lambda x^2$ and differs for cross entropy.
> >
> > 3. My comment on batch normalization was not meant to imply that *'additional constraints by BN may lead to better results'*, but to express doubts on whether a the loss minimizer is a simplex ETF at all (this assumption is made throughout your proofs). Later on in your response you state that 'A simplex ETF structure can be arranged to satisfy batch normalization regardless of the value.'. This is not obvious and should be a theorem/lemma in the paper.
> >
> > ### Experiments
> > *'Our bounds provide the worst-case scenario bound on the intra-class and inter-class cosine similarity. Since we proved our theorems in the appendix, it is not necessary to verify our theorems experimentally. However, we would like to note that the exact intra-class and inter-class cosine similarity resulting in a training process is heavily dependent on the training dynamics and the underlying model architecture, therefore predicting the exact cosine similarity value or its scaling is much more involved and is not within the scope of the paper.'*
> > I strongly disagree. First of all, your theory requires $\epsilon$ to be small enough and it is not clear, whether this regime is achieved in practice. Second, your theory provides bounds that only depend on the achieved training loss value, but not on the optimization dynamics. Third, to make valid conclusions from the theory, it is required that the bounds are somewhat tight, so evaluating the tightness of the bounds is important to ensure that the bounds are not vacuous.

---

> > > ### Author Response · Authors · 2023-11-23
> > > **Reply to reviewer Svsj's comments**
> > >
> > > We thank the reviewer for carefully reading our response and we understand the source of concern of the reviewer regarding our theorems. We also thank the reviewer for providing constructive feedback on our work. Below are our additional comments regarding the reviewer's new response:
> > > 1. Our theorems explicitly stated that $\frac{\epsilon}{\delta} \ll 1$, which is a common notation to imply that higher order terms of $\epsilon$ can be safely ignored, therefore the non-asymptotic bound is a valid bound for our theorems. We understand that the reviewer is mainly concerned with the practicality of the theorems in real neural network training and whether $\frac{\epsilon}{\\delta} \ll 1$ is achieved in practical training. This is indeed a valid concern. To better show the practicality of our theorems, we will conduct additional empirical analysis on the corresponding $\epsilon$ value of our theorems and compare our theorem's predictions with actual empirical results.
> > > 2. Although we're still not fully convinced of the necessity of the adjustments, we will consider and discuss your suggestions more thoroughly and see whether we can adopt the mentioned adjustments to enhance our theorems' robustness.
> > > 3. We would again like to clarify that "the loss minimizer is a simplex ETF" is not an assumption of our theorems but an implication that can be inferred from our theorems by setting $\epsilon=0$ in the non-asymptotic bound. This optimal-case minimizer is not emphasized as a conclusion in our theorems because similar conclusions have been shown in various prior works (e.g. [1][2]) and are not the main contribution of our work.
> > >
> > > ## Experiments
> > > We would like to note that the main purpose of the experiments is to establish that batch normalization and high weight decay improve the proximity to the neural collapse structure, which is sufficiently justified by our experiments. This is an insight that can be derived from our theoretical results while not a strict implication of our theorems.
> > >
> > > However, it would indeed enhance our contributions to empirically evaluate the quality and tightness of our bounds in practical settings and show that near-optimal loss is indeed achieved in our experiments. As we mentioned above, we will perform further analysis of our experimental results and evaluate the tightness of our theorems.
> > >
> > > [1] Zhihui Zhu, Tianyu Ding, Jinxin Zhou, Xiao Li, Chong You, Jeremias Sulam, and Qing Qu. A geometric analysis of neural collapse with unconstrained features. arXiv preprint arXiv:2105.02375, 2021.
> > >
> > > [2] Jianfeng Lu and Stefan Steinerberger. Neural collapse under cross-entropy loss. Applied and Computational Harmonic Analysis, 59:224–241, 2022.

---

> ### Author Response · Authors · 2023-11-22
> **Reply to reviewer Svsj [2/3]**
>
> ## Experiments
>
> - Our bounds provide the worst-case scenario bound on the intra-class and inter-class cosine similarity. Since we proved our theorems in the appendix, it is not necessary to verify our theorems experimentally. I believe the reviewer is referring to verifying how close the actual cosine similarity is to the predicted bound. However, we would like to note that the exact intra-class and inter-class cosine similarity resulting in a training process is heavily dependent on the training dynamics and the underlying model architecture, therefore predicting the exact cosine similarity value or its scaling is much more involved and is not within the scope of the paper. The main insights from the theorems are that 1) Batch normalization and high weight decay imply neural collapse under near-optimal loss and 2) higher weight decay has stronger guarantees of proximity to the Neural Collapse structure. The purpose of our experiments is to investigate these two insights to see whether they transfer into practice.
> - As clarified in the previous point, the purpose of the experiments is not to verify whether the worst-case bounds in our theorems provide a good prediction of the actual intra-class and inter-class cosine similarities, which is heavily dependent on the training dynamics and architecture. Rather, the experiments substantiate the insights gained from the theoretical results, namely batch normalization is an important factor in Neural Collapse Emergence, and higher weight decay results in stronger guarantees of Neural Collapse.
>     1. We thank the reviewer for pointing out that applying batch normalization for every layer is suboptimal. We have rerun our experiments with only last-layer BN along with other changes and the results have indeed shown improved Neural Collapse under our measures.
>     2. We thank the reviewer for pointing out that including training losses will make our plots more informative. We would first like to note that, our theorems do not rely on the unconstrained features assumption so long as near-optimal loss is obtained. On the doubts on the reported intra-class and inter-class results, we think that the following reasons may be influential in the current results for the real-world networks and datasets:
>         1. Achieving near-1 intra-class cosine similarity is a strong requirement on the features within the same class when the vector dimension is large.
>         2. We used the *minimum* intra-class cosine similarity and *maximum* inter-class cosine similarity, which again is a strong requirement for the structure of feature vectors. This measure can be affected by outliers since there are $C(C-1)$ pairs of classes for inter-class cosine similarity
>
>         To address the second point, we would report the average intra-class and inter-class cosine similarity to provide additional measures that are less sensitive to outliers. For the synthetic data experiments, we also adjusted the model architecture to only include batch normalization in the final layer and this has indeed improved the measured results.
>
>     3. We believe that the above 2 replies address this point.
>     4. We thank the reviewer for pointing out this result that seems contradictory to the insights from our theorems. We would acknowledge this phenomenon in our updated version.

---

> ### Author Response · Authors · 2023-11-22
> **Reply to reviewer Svsj [3/3]**
>
> ## General
>
> 1. **Minimal set of conditions for NC.**
>     1. **Motivation.** Past work has demonstrated the emergence of Neural Collapse under various assumptions, such as minimal loss combined with regularization[1], and minimal loss with feature normalization [2][3]. However, there is a significant gap between theory and practice. Empirically, we rarely achieve the optimal loss exactly, nor do we apply regularization and normalization precisely as assumed in these papers. Therefore, the motivation of our paper is to bridge this gap. Specifically, we aim to theoretically identify a minimal set of conditions closely related to practical applications that guarantee the emergence of Neural Collapse.
>     2. **Batch normalization.** The reviewer’s claim that 'The authors argue to add batch normalization to this list but that contradicts minimality' does not accurately reflect our position. Firstly, previous studies have also incorporated various normalization assumptions to facilitate Neural Collapse. Compared to these normalizations, our use of batch normalization and weight decay is more prevalent in practical applications. Therefore, employing batch normalization should not be viewed as an 'additional' assumption, but rather as an alternative or practical implementation of the norm constraint assumption. Secondly, our experiments demonstrate that without batch normalization, Neural Collapse is not significantly observed in some datasets (see Figures 2 and 3). This underscores the necessity of batch normalization from another perspective. Below we outline the role of BN and WD in our theorems in detail:
>         1. To illustrate the above point, our first theorem shows the case where “norm constraint on the features” and near-optimal loss implies Neural Collapse. In this theorem, our theoretical result does not rely on Batch Normalization and only requires the feature norm to be bounded.
>         2. Our second theorem replaces the feature and weight norm constraint with a batch-normalization layer and weight decay. Our results show that replacing the strict norm constraint with a norm penalty is also sufficient for Neural Collapse. Therefore, the BN assumption is not added as an additional assumption to the bounded norm constraint assumption but as a replacement that better resembles practical training.
>     3. **Sub-optimal Loss.** Another area where our work contributes is in generalizing from the optimal scenario to the sub-optimal one. The theorems in previous works assume a strictly optimal loss, which is a much more stringent requirement than a near-optimal loss, the latter being more representative of practical scenarios. Therefore, in this regard, our assumptions are more 'minimal' compared to those in prior theorems. As we clarified previously, our big-O notation is used to simplify the formula by removing constants and higher order terms does not imply that our theorem only holds when $\epsilon\rightarrow 0$.
>     4. **Weight decay.** Furthermore, we have provided a quantitative analysis of how various parameters influence the bound, specifically the gap from optimal loss and the weight decay parameter, aspects not explicitly analyzed in prior works. Weight decay, while widely used in practice, has a sparse theoretical understanding. Our work contributes to understanding weight decay from the perspective of NC.
> 2. We thank multiple reviewers for pointing out the potentially misleading expression of our contributions on the cosine similarity. We are aware that cosine similarity has been widely used in previous literature. Our main claim is that we’re novel in applying quantitative analysis of the cosine similarity measure in the theoretical analysis of Neural Collapse. We will clarify our contributions with regard to this statement and discuss the related use of the cosine similarity measure for similar purposes in previous literature.
>
> We also thank the reviewer for pointing out minor errors in our manuscript and will fix them in our updated version.
>
> ## Questions
>
> 1. A simplex ETF structure can be arranged to satisfy batch normalization regardless of the $\gamma>0$ value. To avoid confusion, we would like to clarify that we consider “scaled simplex ETF”, which can have vector norms to be any value (that is the same across different classes) and is also the case in the original NC literature. Therefore, we do not believe that this affects the validity of our theorems
> 2. As we mentioned in the global response, the big-O notation is mainly intended to omit constants and higher-order terms instead of implying that our bounds only hold when $\epsilon\rightarrow 0$. However, this assumption is still required for our theorem to hold true.

---

> ### Author Response · Authors · 2023-11-22
> **Reply to reviewer Svsj [References]**
>
> [1] Zhihui Zhu, Tianyu Ding, Jinxin Zhou, Xiao Li, Chong You, Jeremias Sulam, and Qing Qu. A geometric analysis of neural collapse with unconstrained features. arXiv preprint arXiv:2105.02375, 2021.
>
> [2] Jianfeng Lu and Stefan Steinerberger. Neural collapse under cross-entropy loss. Applied and Computational Harmonic Analysis, 59:224–241, 2022.
>
> [3] Can Yaras, Peng Wang, Zhihui Zhu, Laura Balzano, and Qing Qu. Neural collapse with normalized features: A geometric analysis over the Riemannian manifold.

---

### Author Response · Authors · 2023-11-20
**Global Response on Asymptotic Notations and updates to Theorem 3.2**

We thank the reviewers for their meticulous review and constructive feedback on our work. In this global response, we address a few suggestions/feedbacks mentioned by multiple reviewers and provide a few updates we have made regarding our results.

Firstly, We carefully examined our proof for theorem 3.2 and found that our results can be improved to remove the exponential dependence on $\frac{1}{\lambda}$. Specifically, for theorem 3.2, we updated our bound to remove the exponential dependency on $\frac{1}{\lambda}$. Assuming
$\epsilon<\lambda$ and under the original assumptions of theorem 3.2:

$\mathit{intra}_c\geq 1-O\left((\frac{C}{\lambda})^{O(C)}\sqrt{\frac{\epsilon}{\delta}}\right)$

$inter_c\geq -\frac{1}{C-1}+O((\frac{C}{\lambda})^{O(C)}(\frac{\epsilon}{\delta})^{1/6})$

Also, several reviewers mentioned that our bounds are asymptotic and cannot be used to compute the exact bound for a given value of $\epsilon$. We would like to clarify that while the bound in theorems is formulated using asymptotic notation, the use of this notation is mainly intended to remove the constants and higher-order terms to simplify the formula in the theorem. It is possible to use our result to generate predictions for given values of $\epsilon$ as long as they are small enough for higher order terms of $\epsilon$ (e.g. $\epsilon^2$) to be safely ignored, which is the case for most successful training of over-parameterized neural networks.  Specifically, for some small constant $\kappa>0$, the non-asymptotic bounds without big-O notation in theorem 3.1 are as follows.

$\mathit{intra}_c\geq 1-\frac{\exp(\kappa C\alpha\beta)}{\alpha\beta}\sqrt{\frac{128\epsilon}{\delta}}$

$inter_{c,c’}\leq -\frac{1}{C-1}+\frac{C}{C-1}\frac{\exp(\kappa C\alpha\beta)}{\alpha\beta}\sqrt{\frac{2\epsilon}{\delta}}+4(\frac{2\exp(\kappa C\alpha\beta)}{\alpha\beta}\sqrt{\frac{2\epsilon}{\delta}})^{1/3}+\sqrt{\frac{\exp(\kappa C\alpha\beta)}{\alpha\beta}\sqrt{\frac{2\epsilon}{\delta}}}$

The bounds without big-O notation for theorem 3.2 can be obtained by substituting $\alpha\beta=\log(C/\lambda)+\sqrt{\frac{C\epsilon}{(C-1)\lambda}}$ in theorem 3.1.

The individual reviews by different reviewers are addressed in the reply to the reviewers.

## Paper Update

We are sincerely grateful for the multiple constructive feedbacks by the reviewers on our paper. Based on the reviewers suggestions, we made many changes to improve our paper. The major changes are outlined below while minor improvements are mentioned in the individual response to the reviewers:

### Experiments:

The reviewers made multiple insightful suggestions on the setup of our experimental evaluations. We accepted the reviewers' suggestions and made the following improvements to our experimental setup to better achieve the near-optimal regime and simulate real-world deep neural network architectures. After the adjustments, we found that the results are significantly more supportive of the insights gained from our theoretical results, namely BN and WD improve the degree of NC. The adjustments we made to our experimental setup are as follows

- For synthetic dataset experiments, we increased the hidden layer sizes from 200 to 300 and increased the smaller model depth from 3 to 4 to guarantee fitting the MLP3 dataset
- For synthetic dataset experiments, we changed the models to only have Batch normalization before the last layer to better represent our theoretical setup and practical models
- For all experiments, the epoch number is increased to 300 (from 200 and 100 respectively), and the optimizer is changed from SGD to Adam
- For real-world dataset experiments, to better compare between BN and non-BN models, we used VGG19 instead of ResNet in the main paper (since there’s not a standard implementation of ResNet without BN layers)

### Main Paper:

- We updated our description of our contributions with regards to the cosine similarity to provide a more accurate reflection of our relationships with prior works
- For both experimental result figures, we combined the BN and non-BN model results into a single figure for better comparison.
- We also plotted **both** the average value and the worst-case value for inter-class and intra-class cosine similarity. We note that the average values are less sensitive to outliers and can provide more information on the trend of the measures with respect to weight decay values
- We updated our definition of the cosine similarity measure in section 2.2 to make them generalizable to models with the bias term in the batch normalization layer

### Appendix

- For both theorems, we added a new bound that represents NC3 (alignment between weight vectors and corresponding class center) measured using cosine similarity
- For all bounds, we added the non-asymptotic version to the appendix theorem. (with the exception of inter-class cosine similarity for theorem 2.2, which is too complex to expand)

---

### Meta-Review · Area_Chair_21wj · 2023-12-15

**Metareview:**

This work delves into a theoretical exploration of the connection between Batch Normalization (BN), weight decay, and a nearly optimal Network Capacity (NC) structure. Similar to previous studies, such as unconstrained models proposed by Zhu et al. (2021), the theoretical analysis primarily focuses on last-layer features, but it approaches them from a near-optimal standpoint. The theoretical analysis reveals that both BN and weight decay play crucial roles in achieving neural collapse in the near-optimal regime.

Despite the interesting theoretical results, the reviewers are majorly concerned about the disconnection between the theoretical and experimental results. For example, (i) the experimental results cannot be fully predicted by the theory, (ii) extension beyond the unconstrained feature models, (ii) lack of experiments on large-scale network architectures.

The work can be improved by incorporating the reviewers' feedbacks, and worth publishing in the near future.

**Justification For Why Not Higher Score:**

Most of the reviewers agree that the work requires significant improvement, and not ready for publication at the current stage

**Justification For Why Not Lower Score:**

N/A

---

### Decision · Program_Chairs · 2024-01-16

Reject